# GAGA zinc finger transcription factor searches chromatin by 1D−3D facilitated diffusion

Xinyu A. Feng[1,2,6], Maryam Yamadi[1], Yiben Fu[2,7], Kaitlin M. Ness[1], Celina Liu[1], Ishtiyaq Ahmed[2,8], Gregory D. Bowman [2], Margaret E. Johnson [2], Taekjip Ha [2,3,4] ✉ & Carl Wu [1,5] ✉

The search for target sites on chromatin by eukaryotic sequence-specific transcription factors (TFs) is integral to the regulation of gene expression but the mechanism of nuclear exploration has remained obscure. Here we use multicolor single-molecule fluorescence resonance energy transfer and single-particle imaging to track the diffusion of purified *Drosophila* GAGA factor (GAF) on DNA and nucleosomes. Monomeric GAF DNA-binding domain (DBD) bearing one zinc finger finds its cognate site through one-dimensional (1D) or three-dimensional (3D) diffusion on bare DNA and rapidly slides back and forth between naturally clustered motifs for seconds before dissociation. Multimeric, full-length GAF also finds clustered motifs on DNA through 1D−3D diffusion but remains locked on target for longer periods. Nucleosome architecture effectively blocks GAF-DBD 1D sliding into the histone core but favors retention of GAF-DBD once it has bound to a solvent-exposed motif through 3D diffusion. Despite the occlusive nature of nucleosomes, 1D−3D facilitated diffusion enables GAF to effectively search for clustered cognate motifs in chromatin, providing a mechanism for navigation to nucleosomal and nucleosome-free sites by a member of the zinc finger TF family.

In eukaryotic organisms, DNA is packaged into chromatin to regulate accessibility and recruitment of DNA-binding proteins involved in DNA replication[1], transcription[2–4] and genome maintenance[5]. Sequence-specific transcription factors (TFs) access cognate DNA motifs on the genome to regulate transcription of target genes. Bacterial TFs such as LacI search efficiently for their targets on the relatively bare bacterial chromosome using a combination of one-dimensional (1D) and three-dimensional (3D) diffusion, referred to as facilitated diffusion[6–10]. How TFs search for their targets on eukaryotic chromatin is less understood because of the presence of nucleosome particles, higher-order chromatin structure, multimerization of TFs and limited experimental investigations on how nucleosomes influence the search process[10].

Chromatin can restrict TF accessibility by positioning nucleosomes over cognate motifs[11], structurally occluding half of the nucleosomal DNA surface and altering DNA conformation. Prior in vitro

¹Department of Biology, Johns Hopkins University, Baltimore, MD, USA. ²Department of Biophysics, Johns Hopkins University, Baltimore, MD, USA. ³Howard Hughes Medical Institute and Program in Cellular and Molecular Medicine, Boston Children's Hospital, Boston, MA, USA. ⁴Department of Pediatrics, Harvard Medical School, Boston, MA, USA. ⁵Department of Molecular Biology and Genetics, Johns Hopkins School of Medicine, Baltimore, MD, USA. ⁶Present address: Bioengineering Department, Stanford University, Stanford, CA, USA. ⁷Present address: School of Biomedical Sciences and Engineering, South China University of Technology, Guangzhou International Campus, Guangzhou, China. ⁸Present address: Department of Microbiology and Biochemistry and Molecular Genetics, Rutgers University NJMS, Newark, NJ, USA. ✉e-mail: taekjip.ha@childrens.harvard.edu; wuc@jhu.edu

studies showed TFs can have unique preferences for binding certain DNA locations on a nucleosome[12–15]. Some TFs can invade nucleosomes by unwrapping DNA at the nucleosome edge or disrupting higher-order chromatin organization[16–18]. Nonetheless, the search mechanisms by which any eukaryotic TF locates its targets, binds selectively to certain nucleosomal sites over others and remains stably associated are poorly understood. Here, we address these questions for *Drosophila melanogaster* GAGA factor (GAF), a single-zinc-finger (ZnF) TF studied extensively using genetic, biochemical, genomic and live-cell, single-molecule imaging approaches[19–26]. GAF is a multifunctional TF that promotes chromatin accessibility at cognate promoter, enhancer and insulator sites. It also has an essential role in promoter–promoter and promoter–enhancer looping, as well as assembly of the transcription preinitiation complex (PIC) with the ensuing paused RNA polymerase II (refs. [27–35]). GAF belongs to the $Cys_2$–$His_2$ ZnF family, which constitutes the largest family of eukaryotic TFs, with ~750 of 1,600 human TFs containing one or multiple ZnF DNA-binding domains (DBDs)[36]. The nuclear magnetic resonance structure of GAF-DBD shows a single ZnF binding to the major groove of the first three bases, GAG, of the GAGAG consensus binding site ('cognate motif'). Two short basic regions (BRs) N-terminal to the ZnF contact the fourth (A) and fifth (G) bases in the minor and major grooves, respectively[24]. Despite its single BR-ZnF domain, full-length GAF (GAF-FL) forms a range of multimeric (on average hexameric to octameric) complexes through the N-terminal poxvirus and zinc finger (POZ) domain[37] and preferentially binds to clusters of cognate motifs[26,32,38] (Fig. 1a).

Here, we used multicolor single-molecule fluorescence resonance energy transfer (smFRET)[39,40] to study the target search process of GAF using purified components. Using FRET as a readout for cognate-motif-specific binding, we found that the DNA sequence specificity of GAF-DBD is kinetically defined by rates of both DNA association and DNA dissociation. During target search, GAF-DBD undergoes 1D diffusion on free DNA with transient residence at cognate motifs. While remaining on the DNA, GAF-DBD frequently escapes from a motif to locate a neighboring cognate site. Sites at the edge of a nucleosome can be located by 1D invasion from linker DNA. By contrast, inner nucleosomal motifs require direct 3D association from solution; however, once bound, dissociation is slower than from the edge. Using optical tweezers coupled with confocal microscopy, we also show that GAF-FL multimers slide on DNA and form stable complexes with naturally clustered cognate sites. Together, our results provide mechanistic insights on how a eukaryotic TF combines 1D and 3D diffusion to search for and locate its target sites on free DNA and nucleosomes.

## Results

### Visualizing sequence-specific DNA binding by smFRET
To investigate how GAF searches for its cognate motif on DNA, we placed a FRET donor (Cy3) at the N terminus of GAF-DBD (using one-pot N-terminal cysteine labeling[41]; Extended Data Fig. 1a) and a FRET acceptor (Cy5) on the DNA, 5 bp from a GAGAGAG sequence consisting of two overlapping GAGAG motifs (Fig. 1b, first inset). The 90-bp DNA is biotinylated on one end and immobilized for smFRET imaging (Fig. 1b, second inset). When GAF-DBD is specifically bound to the motif, the donor–acceptor proximity results in acceptor emission by FRET (Fig. 1b, third inset). When it is nonspecifically bound, the greater average distance results in primarily donor emission (Fig. 1b, fourth inset).

### GAF-DBD locates cognate motif on linear DNA by 1D sliding
Single GAF-DBD binding to surface-immobilized DNA results in an abrupt appearance of Cy3 fluorescence above background (Fig. 1c). In an example single-molecule fluorescence time trajectory, a GAF-DBD molecule associates with the cognate motif at ~9 s showing high FRET and dissociates at 14 s; another molecule binds at the cognate site again at ~31 s and quickly dissociates; two binding events at ~48 s and 78 s show Cy3 donor emission only, attributed to nonspecific binding.

GAF-DBD successfully locates the cognate motif 68% ± 5% of the time (534 events analyzed). This success rate is much higher than 2%—the expected probability if target search solely relies on random 3D collisions until the target motif is hit (calculated by dividing the number of GAGAG cognate binding sites by the total number of 5-mer sites). Thus, we hypothesize that, after initial 3D binding, GAF-DBD may slide on DNA to search for the cognate target (hereafter, sliding is used interchangeably with 1D diffusion, which includes both helically coupled 1D sliding and 1D hopping). Accordingly, a time lag between the appearance of donor and ensuing acceptor fluorescence would represent the period between GAF-DBD landing anywhere on DNA and the successful recognition of a cognate site by sliding. Such time delays are observed for 20% of the binding events (Fig. 1d). The remaining events may also be preceded by a sliding period that is too transient to be detected under our instrument resolution (100 ms). In addition, we observe spikes of Cy3 donor-only fluorescence within FRET events, which suggests that GAF-DBD transiently slides off the target before quickly returning (Fig. 1e, asterisks). At lower ionic strength, the nonspecific dwell time increases, suggesting that 1D diffusion involves some 1D hopping along the DNA[42] (Extended Data Fig. 2a–d); this is consistent with some TFs shown to lose sequence specificity in low-salt conditions[43].

### DNA-binding kinetics define sequence specificity of GAF-DBD
When we replaced the GAGAGAG motif with a non-target sequence TATACAG, we observed only transient Cy3 donor emissions (Fig. 1f), further establishing FRET appearance as a readout of motif-specific binding. Moreover, high labeling efficiency for GAF-DBD (92%) allows reliable counting of DNA-binding events. We find that GAF-DBD's overall dwell time (regardless of being diffusive or immobile) on noncognate DNA ($1/k_{off}$ = 0.33 s) is ninefold smaller than on cognate DNA (3.1 s) (Fig. 1h and Extended Data Fig. 2e). At 0.2 nM protein concentration, the detectable binding frequency on noncognate DNA (0.0018 $s^{-1}$ or one binding attempt every 560 s) is 16-fold lower than on cognate DNA (0.028 $s^{-1}$ or one binding attempt every 36 s) (Fig. 1g). Although nonspecific binding events shorter than the camera exposure time (100 ms) would go undetected, the latter finding suggests that the apparent association rate is increased by the presence of the GAGAG cognate sequence. The calculated $K_D$ (based on single-molecule $k_{off}$ and $k_{on}$ measured for cognate binding events under different protein concentrations) agrees with the $K_{1/2}$ of ~12 nM measured using an electrophoretic mobility shift assay (EMSA) (Extended Data Fig. 1b,c). Our results indicate that the cognate site both increases detectable binding frequency ($k_{on}$) and transiently traps GAF-DBD to reduce dissociation ($k_{off}$) to achieve high sequence specificity ($K_D$), consistent with recent findings for other TFs[16,17,44,45].

### Three-color FRET reveals back-and-forth sliding of GAF-DBD
On the *Drosophila* genome, GAF preferentially binds to closely spaced cognate motifs at numerous promoters and enhancers[38]. To mimic a native GAF target, we used a 187-bp segment from the *Drosophila hsp70* promoter sequence, retaining the two longest GAF-binding sites while mutating several other short GA elements (Fig. 2a). Two FRET acceptors, Cy7 and Cy5, were placed 3 bp away from site 1 (GAGAGGGAGAGA) and 5 bp away from site 2 (GAGAGAG), respectively, with 57 bp of intervening DNA such that FRET to either acceptor signified Cy3–GAF-DBD binding to the corresponding site (Fig. 2a).

In the single-molecule trajectory of Fig. 2b (expanded in Fig. 2d), a GAF-DBD landed on the DNA at ~17 s, with Cy3 signal. Almost immediately after, we observed Cy3–Cy7 FRET, indicating that GAF-DBD located site 1. Cy7 intensity dropped and Cy5 intensity increased 0.25 s later, indicating that GAF-DBD slid away from site 1 to reach site 2. Another 0.25 s later, GAF-DBD slid back to reengage with site 1. Intervening periods of Cy3 fluorescence with neither Cy5 nor Cy7 FRET were observed, indicating nonspecific binding during transit.

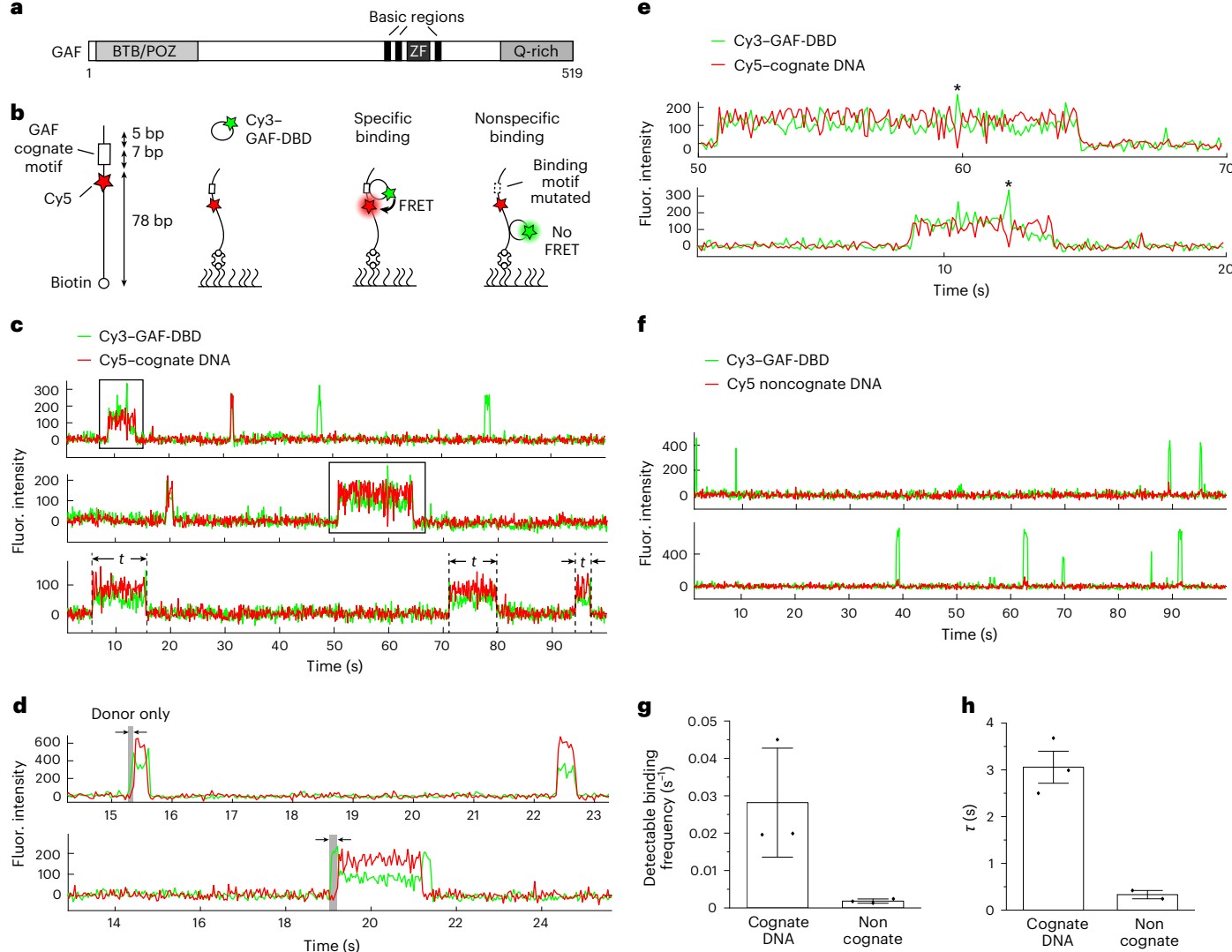

**Fig. 1 | DNA sequence specificity of GAF-DBD is kinetically defined. a**, Domain map of GAF. ZF, zinc finger. **b**, Schematics of two-color smFRET experiment to measure cognate-specific binding of GAF-DBD. **c**, Single-molecule trajectories showing Cy3–GAF-DBD binding to Cy5–DNA containing a GAF cognate site. Boxed binding events are magnified in **e**. Examples of dwell time measurements (*t*) are shown on the third trajectory. **d**, Representative single-molecule trajectories of Cy5–cognate DNA bound by Cy3–GAF-DBD showing an initial donor-only period before acceptor signal increases. The donor-only period is highlighted in gray. **e**, Zoomed-in view of binding events boxed in **c**. The asterisk indicates a transient Cy3-only fluorescence spike during binding. **f**, Single-molecule trajectories showing Cy3–GAF-DBD nonspecifically binding to Cy5–DNA where the cognate site was substituted with a noncognate sequence. **g**, Binding frequency of GAF-DBD to cognate or noncognate DNA. **h**, Dwell time of GAF-DBD on cognate or noncognate DNA. Error bars show the s.e.m. from mean binding frequencies of technical replicates (cognate DNA, three replicates, total *n* = 317; noncognate DNA, two replicates, total *n* = 282). The concentration of GAF-DBD used in these experiments was 0.2 nM. Fluor., fluorescence.

To automate binding site assignment, we developed a hidden Markov model (HMM) that uses three-color fluorescence intensities to infer the most likely sequence of GAF-DBD binding positions (Supplementary Note). The results are plotted as colored ribbons where blue segments represent site 1 binding, red segments represent site 2 binding and green segments represent nonspecific binding (Fig. 2b). For the 50 randomly selected dwells on DNA shown, blue and red segments are interspersed with green, indicating that GAF-DBD undergoes 1D diffusion for every specific binding event (Fig. 2c). Note that if the transition time between site 1 and site 2 is shorter than the camera exposure (35 ms), the ribbon may appear as a directly concatenated blue–red segment without green intervention (Fig. 2d, 17.9 s). Such 'direct transitions' account for ~50% of 573 transitions. On average, dwell times on site 1 (0.25 s) are longer than on site 2 (0.11 s) (Fig. 2e), likely because of its longer cognate sequence (12 bp versus 7 bp). The overall dwell time on this DNA fragment is 3.34 s, similar to the 3-s

dwell time on the 90-bp DNA harboring a single site identical to site 2 (Fig. 1g). When site 1, marked by Cy7, was mutated, Cy3–Cy7 FRET became transient or undetectable while Cy3–Cy5 FRET remained stable (Extended Data Fig. 3c–e), further supporting our interpretation of 1D sliding to locate a motif. We conclude that 1D diffusion by GAF-DBD on DNA is an important and integral part of its functional search for a specific DNA target (Fig. 2f).

On the natural *hsp70* promoter, site 1 and site 2 are located on different strands of the DNA. To reflect the natural orientation, we replaced site 1 with the complementary sequence 'CTCTCCCTCTCT'. Thus, the BR-ZnF domain must flip 180° orthogonal to the DNA axis to recognize the other site. Interestingly, GAF-DBD still exhibits similar FRET dynamics (Extended Data Fig. 3f–h) and a similar overall dwell time on DNA (3.6 s versus 3.34 s), indicating that GAF-DBD can readily flip on DNA during 1D diffusion to sample different motif orientations.

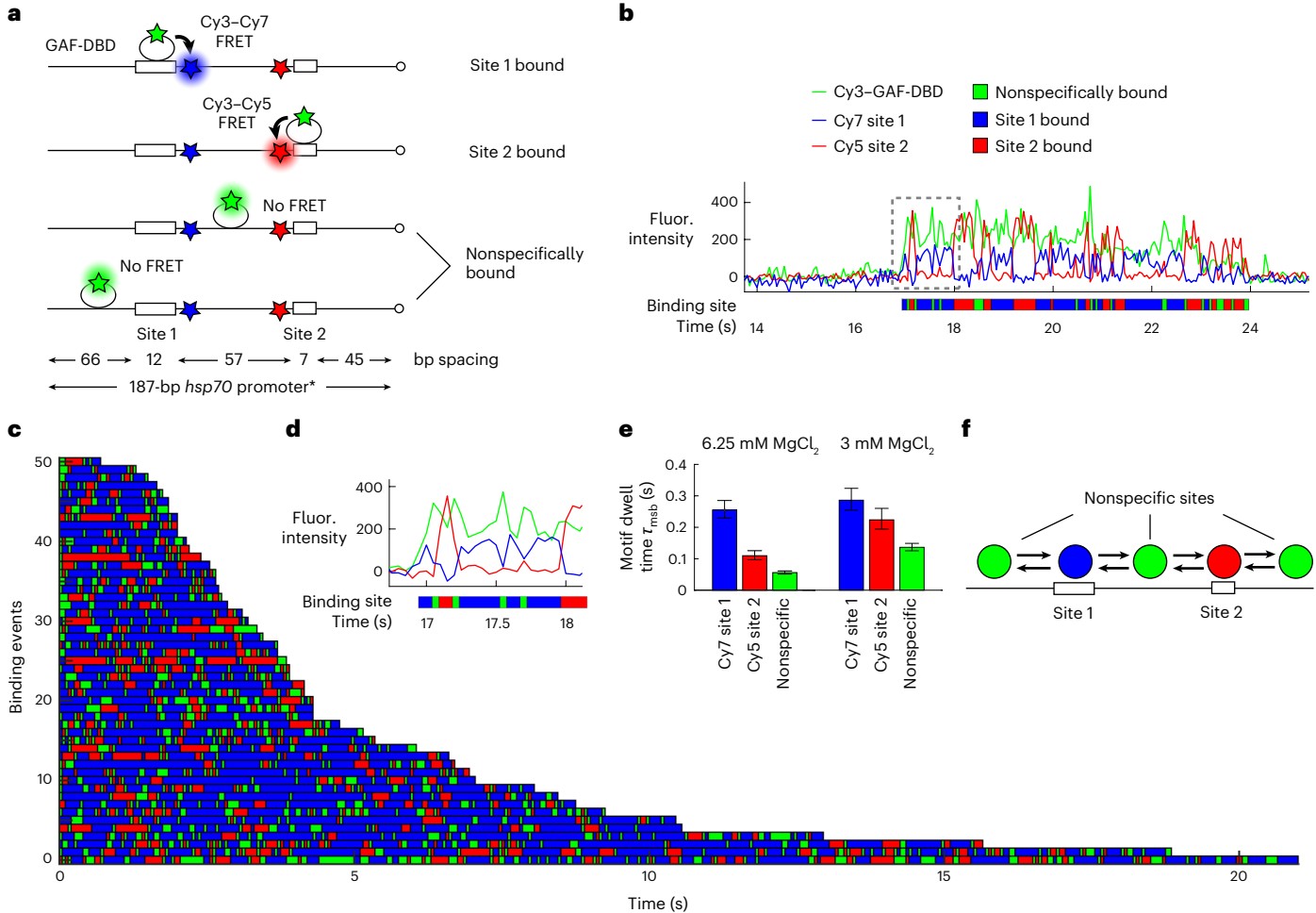

**Fig. 2 | GAF-DBD explores free DNA by 1D diffusion. a**, Three-color smFRET experiment distinguishes whether GAF-DBD is bound to Cy7-labeled site 1, Cy5-labeled site 2 or a nonspecific site on *Drosophila hsp70* promoter DNA. **b**, A single-molecule trajectory shows GAF-DBD sliding back and forth on the DNA between two cognate motifs. Binding site assignment is shown below as a colored ribbon. **c**, A total of 50 binding events shown as a rastergram. **d**, Zoomed-in view of the boxed region in **b**. **e**, GAF-DBD dwell times for motif-specific binding ($\tau_{msb}$)

on site 1, site 2 or nonspecific site on the DNA with regular (6.25 mM, $n = 51$ traces) or low (3 mM, $n = 32$ traces) MgCl$_2$. The concentration of GAF-DBD used in these experiments was 0.1 nM. Error bars represent 95% confidence intervals. **f**, Schematic of GAF-DBD sliding on DNA. Circles represent GAF-DBD. Green circles, GAF-DBD binding to a nonspecific site; blue circle, GAF-DBD binding to site 1; red circle, GAF-DBD binding to site 2.

## GAF-DBD also directly binds to sites by 3D diffusion

Next, we wondered whether every specific binding event is preceded by 1D sliding, identified by a delay between nonspecific DNA binding (Cy3-only fluorescence) and site recognition (FRET). Such a delay is observed for 40–50% of binding events; the remainder displays FRET immediately upon GAF-DBD landing, binding by 3D diffusion (Extended Data Fig. 3i) or undetected 1D diffusion from a nonspecific site. It is unlikely that the latter accounts for all remaining events because, at a lower ionic strength (3 mM versus 6.25 mM MgCl$_2$), GAF-DBD exhibits a threefold longer dwell time at nonspecific sites (Fig. 2e and Extended Data Fig. 3a,b), yet we still observe FRET 39% of the time upon binding (Extended Data Fig. 3i), suggesting that a notable fraction of immediate FRET-upon-binding events reflect direct binding from 3D.

## Nucleosome impedes 1D sliding into its core

We next tested whether GAF-DBD uses 1D diffusion to find cognate motifs within a nucleosome using three 40-N-40 mononucleosome constructs. Each nucleosome contained a Cy5-labeled GAGAGA site on the linker DNA and a second Cy7-labeled GAGAGA site at distinct superhelical locations (SHLs), SHL7, SHL5 or SHL3, inside a nucleosome positioned by the Widom 601 sequence[46] (the major groove of

each cognate motif facing the histone core) (Fig. 3a and Extended Data Fig. 4a). For the nucleosome bearing a motif at SHL7, which is at the nucleosome edge, alternating Cy5 and Cy7 fluorescence is observed for 75% of 485 binding events, indicating 1D sliding between linker and SHL7 sites (Fig. 3b,e). The remaining binding events show either stable residence on linker DNA (19%) or residence on SHL7 site only (7%) (Fig. 3h). In contrast, cognate sites placed at more internal locations, SHL5 or SHL3, are rarely bound; ~90% of binding events show Cy3–Cy5 FRET, suggesting that linker DNA site binding dominates over SHL5 and SHL3 site binding (Fig. 3c,d,f–h).

To narrow down the nucleosome boundary that blocks 1D sliding, we measured GAF binding to a cognate site at SHL6.5. Binding to this nucleosome resembles SHL3 site and SHL5 site nucleosomes, with much fewer sliding events (4.5%) and more binding to linker site only (83%) compared to the SHL7 site construct (67% sliding and 20% linker site binding only) (Extended Data Fig. 4b,c). These results indicate nucleosome organization allows 1D sliding of GAF-DBD to no farther than its outer edge, effectively blocking sliding even 5 bp deeper into the nucleosome core. Thus, transient, intrinsic DNA-unwrapping dynamics[47] is apparently insufficient for 1D sliding past SHL7 into the nucleosome core.

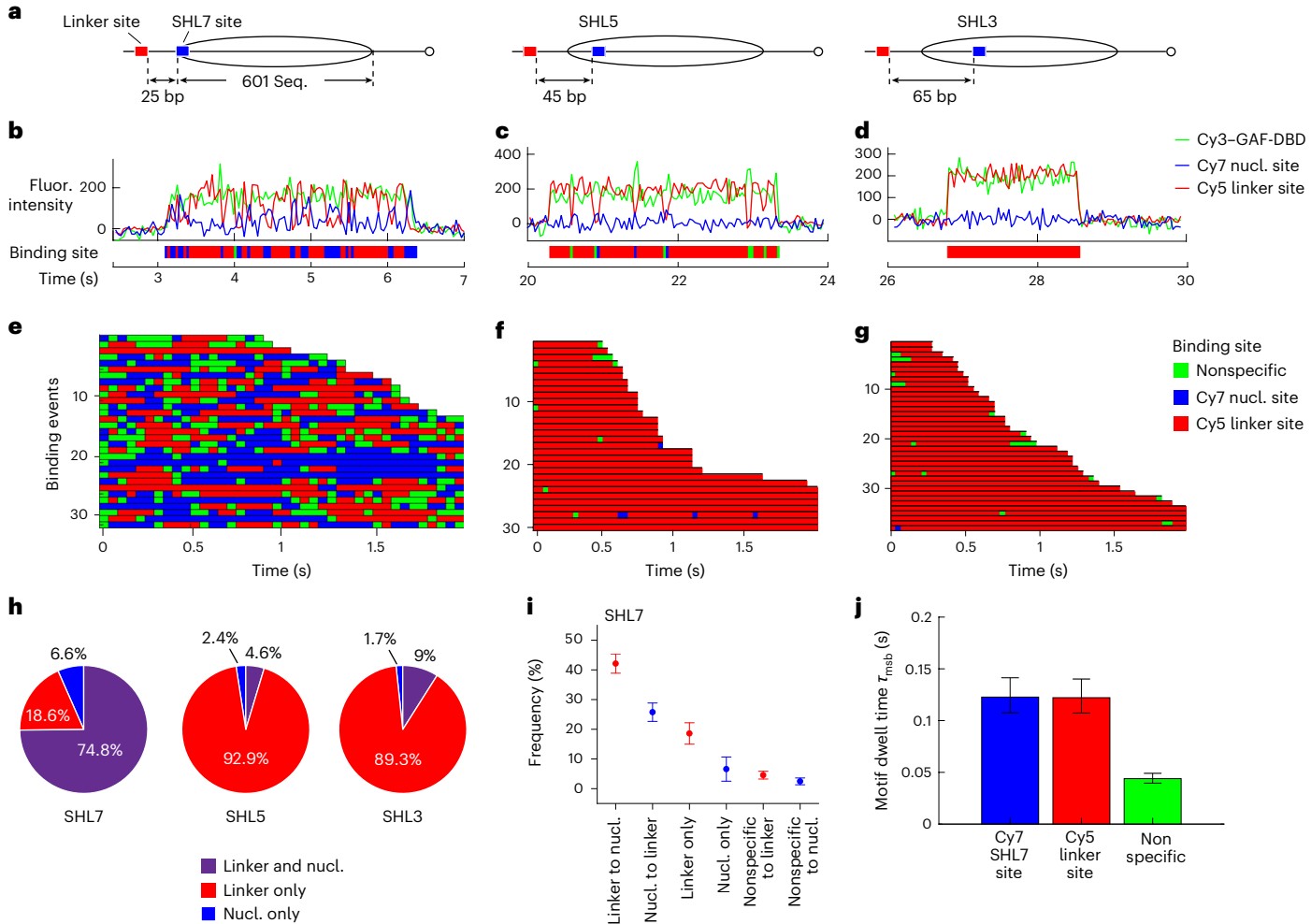

**Fig. 3 | Nucleosome blocks GAF-DBD 1D sliding beyond SHL7. a,** Schematics of nucleosome constructs where the nucleosomal Cy7-labeled cognate site (blue) is placed at SHL7, SHL5 or SHL3 and the other Cy5-labeled cognate site (red) is on the linker DNA. The nucleosome is positioned by the Widom 601 sequence flanked by 40-bp linker DNA on both sides (40-N-40). Seq., sequence. **b–d,** Representative single-molecule trajectories of GAF-DBD binding to linker or nucleosomal sites at SHL7 (**b**), SHL5 (**c**) and SHL3 (**d**). **e–g,** Binding site rastergrams for GAF-DBD on nucleosome constructs where the Cy7-labeled motif

is located at SHL7 (**e**), SHL5 (**f**) or SHL3 (**g**). **h,** GAF-DBD binding categories for each construct; $n = 485$ for SHL7, $n = 260$ for SHL5 and $n = 272$ for SHL3. **i,** Categories of binding events on SHL7 construct. Data are represented as the mean values ± s.d. of three independent imaging sessions, with a sum of $n = 485$ binding events. **j,** GAF-DBD motif dwell times on SHL7, linker site and nonspecific sites on the SHL7 construct ($n = 34$ traces). Error bars represent 95% confidence intervals. Nucl., nucleosome.

To bypass this blockade, GAF-DBD may bind directly through 3D diffusion to solvent-exposed nucleosome locations. We shifted the internal cognate sites by 5 bp each to SHL4.5 and SHL2.5 to face the solvent, expecting a higher fraction of direct 3D binding to the nucleosomal site (stable Cy3–Cy7 FRET throughout binding). Indeed, the SHL4.5 site exhibits a 12-fold higher fraction (23%) of nucleosome binding through 3D diffusion than the SHL5 site (2%) and the SHL2.5 site exhibits a threefold increase (21%) compared to SHL3 (7%) (Extended Data Fig. 4b–d). These results demonstrate that solvent-facing cognate motifs are indeed more commonly accessed by 3D diffusion.

**Naked DNA is the preferred landing site over nucleosome core**
On the 601 nucleosome bearing a cognate site at SHL7, more than 60% of the binding events show alternating Cy3–Cy5 and Cy3–Cy7 FRET, characteristic of 1D diffusion (Fig. 3i). Among these sliding events, 62% initially land on the 40-bp linker DNA, compared to 38% directly landing on SHL7, indicating 1.6-fold greater accessibility (association rate) for a site on the linker DNA than on the nucleosome edge. Binding to the linker DNA site only, showing continuous Cy3–Cy5 FRET, is less common (for example, Fig. 3e, row 3, constant red) but is still

more frequent than binding to the SHL7 site only, showing continuous Cy3–Cy7 FRET (constant blue), a further indication that the linker site is more accessible (Fig. 3i). Dwell times on the linker (0.12 s) and SHL7 (0.12 s) sites are similar (Fig. 3j). Hence, location of a cognate site on the nucleosome edge mildly reduces the GAF-DBD on rate but does not affect the off rate.

**3D diffusion enables nucleosome target search**
On native promoters such as *Drosophila hsp70*, several GAGA elements over multiple phases along the DNA helical axis[23] could potentially increase 3D accessibility when wrapped in a nucleosome. Accordingly, we designed and constructed by linker ligation[48], a 0-N-40 *hsp70* promoter nucleosome whose position (−209 to −63 relative to the transcription start site) mimics one of three thermodynamically favored nucleosomes on *hsp70* promoter DNA harboring two native GAGA elements[49,50]. As shown in Fig. 4a, the CTCTCCCTCTCT site in its native orientation is located near the nucleosome dyad and labeled with Cy7; the other site, GAGAGAG, is positioned at the nucleosome edge and labeled with Cy5. We hypothesized that GAF-DBD should locate the nucleosome dyad site predominantly through 3D diffusion, whereas

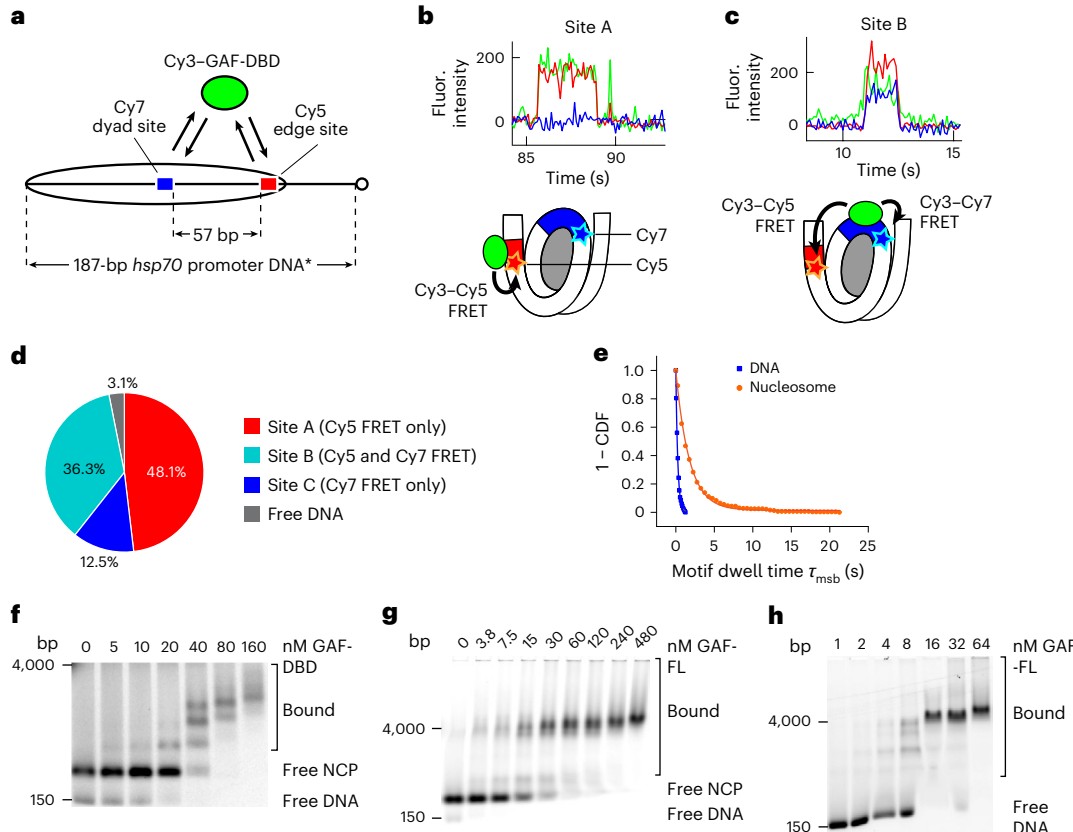

**Fig. 4 | 3D diffusion dominates for inner nucleosomal targets. a**, *Drosophila hsp70* promoter nucleosome construct (0-N-40, same DNA as in Extended Data Fig. 3f) for investigating GAF-DBD search when two binding sites lie within the nucleosome core. **b,c**, Representative single-molecule trajectories for site A (**b**) and site B (**c**) binding (0.4 nM GAF-DBD used). **d**, Binding event categories on the nucleosome. **e**, A 1 − CDF plot of motif dwell times for nucleosome (orange) compared to free DNA (blue). Fitted τ values are provided in Supplementary

Table 5. Free DNA data are aggregated from Cy7–dyad site and Cy5–edge site dwell times on the same DNA template (Extended Data Fig. 3f). **f**, EMSA for GAF-DBD binding to Cy5–*hsp70* NCP. **g**, EMSA showing GAF-FL binding to Cy5–*hsp70* NCP. **h**, EMSA showing GAF-FL binding to Cy5–*hsp70* DNA (the same DNA for Cy5–*hsp70* NCP). EMSA experiments in **f**–**h** were independently repeated at least once with similar results.

1D or 3D diffusion should apply for the nucleosome edge. Three types of binding events are observed (*n* = 160): (1) GAF-DBD binding to the edge (site A; 48%) (Fig. 4b); (2) GAF-DBD binding to the dyad site distal to Cy7 (site B), with FRET to both Cy5 and Cy7 acceptors (36%) (Fig. 4c); and (3) GAF-DBD binding to another internal site (site C), with FRET to Cy7 only (13%) (Extended Data Fig. 5a). Residual binding to unreconstituted free DNA, showing alternating FRET to Cy5 or Cy7, is minimal (3%). Remarkably, 97% of nucleosomal binding events reveal no anticorrelated changes in Cy5 and Cy7 FRET, in sharp contrast to binding to free DNA control (Extended Data Fig. 3f–h), indicating that GAF-DBD remains bound to one site without sliding or hopping to the other site. Together, our results suggest that 3D diffusion is the major search mode for locating inner nucleosomal motifs.

Interestingly, individual nucleosomes exhibit sustained preference for one 3D binding location over others (Extended Data Fig. 5b). For binding events on site A, 88% of the following binding events occur at the same site (type A–A) (Extended Data Fig. 5c). The same rebinding preference is observed for sites B (84%) and C (88%), suggesting that the reconstituted *hsp70* nucleosome population has at least three distinctly phased, stable configurations, each presenting a different preferred site for 3D binding (Extended Data Fig. 5d). We sporadically observe transient Cy3 fluorescence without FRET within site-specific binding on the nucleosome (Extended Data Fig. 5b, black asterisks), suggesting that GAF-DBD may undergo ultrashort-range (<10 bp) 1D diffusion within the exposed cognate site. Strikingly, the average dwell time of GAF-DBD on the two nucleosomal motifs is sevenfold

greater than on naked DNA motifs (Fig. 4e), indicating that nucleosome architecture can better trap GAF-DBD once it binds cognate sites, thus providing some compensation for the nucleosomal impediment to 1D and 3D association. Consistent with this idea, the average GAF-DBD dwell time on solvent-facing motifs at SHL2.5, SHL4.5 and SHL6.5 on a 601 nucleosome is 0.8–0.9 s, eightfold longer than when a similar motif (Cy5 site 2 on *hsp70* DNA) is on naked DNA.

Next, we asked whether the direct binding to a solvent exposed motif on a nucleosome surface we observed for GAF-DBD also applies to GAF-FL (Extended Data Fig. 1d). To test this with EMSA, we first verified that a gel shift can be observed for GAF-DBD binding to an hsp70 nucleosome core particle (NCP; 3-N-6) (Fig. 4f). GAF-FL binds less well to the NCP than to bare DNA but only by ~2-fold (compare bound fractions at ~15 nM protein in Fig. 4g,h). Because the reconstituted NCP contains essentially no linker DNA for 1D sliding, we infer that the cognate sites must be found by 3D diffusion for GAF-FL.

### GAF-FL displays 1D sliding before locking on target
To investigate whether GAF-FL also undergoes 1D diffusion on long DNA, we tracked the position of SNAP-tagged GAF-FL labeled with AlexaFluor488 (AF488) (Extended Data Fig. 6a–c) on kbp-long DNA stretched between two optical traps (C-Trap, LUMICKS) (Fig. 5b and Extended Data Fig. 7c). This DNA is a concatenated 3.3-kbp plasmid carrying, near its edge, a 359-bp insert containing the *hsp70* promoter harboring a dense cluster of full and partial GAGAG sequences (vector + *hsp70*) (Fig. 5a and Extended Data Fig. 7a,b). Accordingly, adjacent

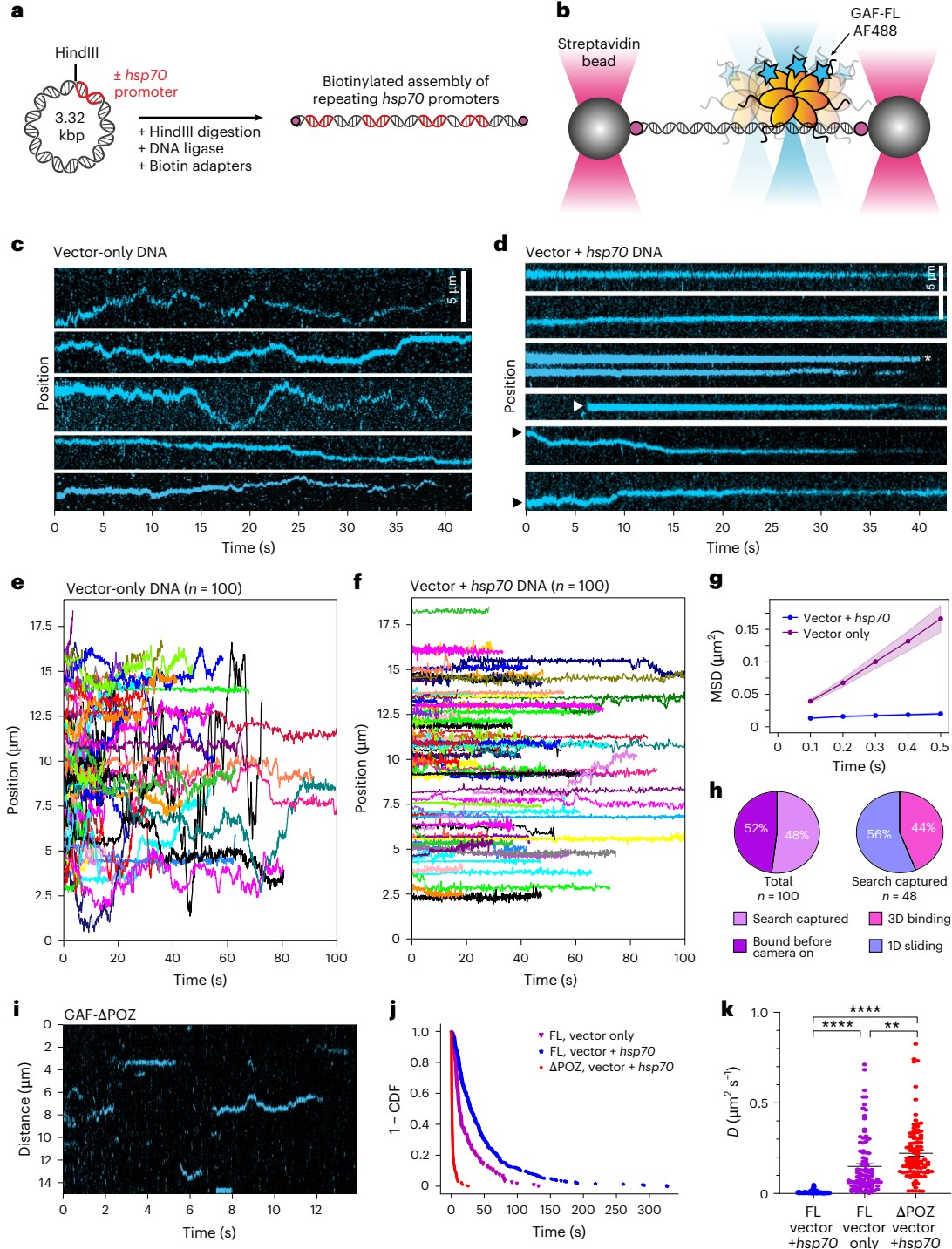

**Fig. 5 | GAF-FL multimer undergoes 1D diffusion during target search.**
**a**, Schematic of construct design. Plasmids with or without the *hsp70* promoter
sequence were digested with HindIII and concatenated using T4 DNA ligase.
Exposed ends were then biotinylated with adaptors, resulting in dual-end
biotinylated DNA carrying repeating *hsp70* promoters. **b**, Setup of dual optical
tweezers for confocal microscopy of stretched DNA tethered between two
streptavidin-coated polystyrene beads. **c**,**d**, Representative kymographs
show AF488–GAF-FL fluorescence signal (cyan) on DNA over time in the
absence (**c**; vector only) or presence of *hsp70* promoter (**d**; vector + *hsp70*).
The white arrowhead shows GAF-FL binding directly to target in 3D; the black
arrowheads show GAF-FL undergoing 1D search before finding target; the white
asterisk shows direct 3D dissociation. Data are representative of at least three
independent imaging sessions. **e**, Compiled plots of position versus time for GAF-
FL trajectories on vector-only DNA. A total of 100 traces were collected for each

condition and arranged to start from $t = 0$. **f**, Compiled plot for GAF-FL positions
over time on vector + *hsp70* DNA. **g**, Average MSD over time lag of all collected
traces for vector only (purple) and vector + *hsp70* DNA (blue). The shaded area
reports the s.e.m. ($n = 100$). **h**, Pie charts categorizing GAF-FL traces by DNA
binding at onset of video (left) and targeting directly (3D binding) or indirectly
(1D sliding) (right). **i**, A representative kymograph of GAF-ΔPOZ on vector + *hsp70*
DNA. **j**, A $1 - $CDF plot comparing dwell times of GAF-ΔPOZ on vector + *hsp70*
DNA (red; $\tau = 1.67$ s), GAF-FL on vector + *hsp70* (blue; $\tau = 43.4$ s) and GAF-FL on
vector only (purple; $\tau = 21.0$ s). **k**, Single-trace diffusion coefficients for GAF-FL
on vector + *hsp70* and vector-only DNA and GAF-ΔPOZ on vector + *hsp70* DNA.
Error bars represent the s.e.m. Statistical differences were determined using
a two-tailed unpaired *t*-test with Welch's correction ($n = 100$ GAF molecules;
**$P = 0.0012$ and ****$P < 0.0001$). These experiments were conducted using 80 nM
GAF-FL and 25 nM GAF-ΔPOZ.

promoters can be zero, one or two plasmid lengths apart, giving rise to a semirandom distribution of *hsp70* promoters over tens of kbp of DNA. Mass photometry analysis of this GAF-FL construct suggests that the purified protein can form a wide range of multimeric states, with the dominant species shifting from monomers to dimers, trimers, tetramers, hexamers and octamers as the protein concentration increases (Extended Data Fig. 8d). At the 80 nM concentration used for tracking, the major species is a GAF-FL octamer.

On vector-only DNA, GAF-FL displays rapid, random 1D diffusion with a mean $D_{coef}$ = 0.15 μm² s⁻¹ or 1,606 bp² s⁻¹, often traversing a few micrometers (~10 kbp) per minute (Fig. 5c,e). Only a very small fraction (4 of 100 binding events) show nondiffusive binding ($D_{coef}$ < 0.01 μm² s⁻¹). In contrast, on vector + *hsp70* DNA, GAF-FL multimers are mostly (78 of 100) nondiffusive and remain immobile for at least 40 s before photobleaching or dissociation (Fig. 5d,f). The different mobilities with and without *hsp70* are also seen in the average mean squared displacement (MSD) over time lag (Fig. 5g). Although a substantial fraction of GAF-FL becomes immobilized on target between protein injection and imaging start (~30 s), de novo binding events are still observed, 44% (21 of 48) of which appear abruptly and remain immobile (Fig. 5d,h, white arrowhead, and Extended Data Fig. 7e) and 56% (27 of 48) exhibit detectable 1D diffusion before stasis, consistent with a 1D search after nonspecific binding (Fig. 5d, black arrowhead, and Extended Data Fig. 7d). Note that the fraction of 3D binding events is likely overestimated given that a transient leading 1D diffusion period may go undetected because of instrument time resolution (70–200 ms) and fast protein diffusion (Methods). We conclude that 1D diffusion is the dominant search mode for GAF-FL on linear DNA. GAF-FL dissociation from target can also occur through 1D (Extended Data Fig. 7f) or 3D (Fig. 5d, white asterisk) diffusion.

The 1D $D_{coef}$ for GAF-FL on vector DNA exhibits a wide distribution (Fig. 5k), likely because of heterogeneity in multimeric states, as observed by mass photometry (Extended Data Fig. 8d), as well as in the number of engaged DBDs for each multimer. The gradual loss of AF488 fluorescence from photobleaching further supports the multimeric state of GAF-FL (Extended Data Fig. 8c). Mass photometry on GAF with the POZ domain deleted (GAF-ΔPOZ, which retains the DBD and Q-rich region) (Extended Data Figs. 1d and 6d,e) shows a single peak at the monomer mass (Extended Data Fig. 8d, red) and its diffusion coefficient (0.222 μm² s⁻¹) is ~30-fold larger relative to GAF-FL (0.007 μm² s⁻¹) (Fig. 5k), higher than the maximum fourfold difference expected from the tenfold size difference between an average GAF-FL multimer (~800 kDa) and ~80-kDa GAF-ΔPOZ. Strikingly, binding of monomeric GAF-ΔPOZ on vector + *hsp70* DNA is much more transient ($τ$ = 1.67 s) (Fig. 5i,j and Extended Data Fig. 8a,b) compared to GAF-FL ($τ$ = 43 s) (Fig. 5j). This transience is similar to the 3–4-s dwell time of monomeric GAF-DBD on short motif-carrying DNA fragments as measured by smFRET, providing in vitro evidence that multimerization through the POZ domain is necessary for prolonged site-specific binding at timescales approaching values measured in living cells (~2 min)[26]. In summary, our results show that specific and prolonged GAF-FL binding on a physiological target requires both the natural cognate site cluster on DNA and GAF protein multimerization.

## Discussion

A key finding of our work is that eukaryotic TFs locate their cognate DNA targets through a combination of 1D and 3D diffusion, including in the presence of nucleosomes. Using multicolor smFRET, we explored whether a *Drosophila* TF, GAF, relies on 1D diffusion for target search on free DNA and on a nucleosomal template, in which histone–DNA interactions may present a barrier to 1D sliding. Similar to LacI and other DNA-binding proteins[9,51–57], we find that 1D diffusion is the major search mode for GAF-DBD to locate its target on nucleosome-free DNA but, notably, the sliding occurs back and forth over 100–200 bp on a subsecond timescale (four orders of magnitude faster than the LacI

dimer switching between adjacent operators[9]), possibly enabled by contacts between the DBD and the DNA sugar-phosphate backbone[24]. In addition, 1D sliding can penetrate into the extreme edge (SHL7) of nucleosomes but not beyond SHL6.5 (Fig. 6a). Using optical tweezers coupled with single-particle imaging, we further show that multimeric GAF-FL also uses 1D diffusion to search for natural promoter targets on naked DNA (Fig. 6b). The POZ multimerization domain is required for the substantially more stable interaction with DNA and, once the target motif cluster has been located, GAF-FL remains on site for about 1 min, nearing physiological timescales[26,58] and comparable to the LacI dimer. Our results not only explain the role of GAF multimerization and the natural clustering of GAGAG elements near GAF-dependent promoters and enhancers[37,38] but also potentially explain why almost all ZnF TFs contain multiple ZnF domains in tandem[36] and why TF affinity to cognate sites can be influenced by surrounding sequences[59]. Our findings indicate 1D sliding at nucleosome-free regions (NFRs) could be a search mode for eukaryotic TFs in general, within and beyond the ZnF family.

Despite the high efficiency of 1D facilitated diffusion for locating bare DNA targets, cognate motifs at SHL6.5 or deeper into the nucleosome core cannot be located using the 1D search mode. A search mode by 3D diffusion allows GAF to find solvent-facing (not core histone-facing) motifs by direct association from solution (Fig. 6a). On a reconstituted *hsp70* promoter nucleosome, the phasing of cognate sites is such that the major groove of at least one motif faces solvent[13,15], allowing 3D targeting. In addition, native (non-601) DNA sequences are intrinsically diffusive on the histone octamer on a single-bp scale, which allows different rotational phases to be sampled for better motif exposure[60,61]. Our optical trap measurements show that multimeric GAF-FL, which contains multiple DBDs, can explore kbp of naked DNA using 1D diffusion. In good agreement with GAF-DBD findings, GAF-FL also locates a substantial portion of binding sites directly through 3D diffusion (although a leading short-range 1D sliding period could be missed, resulting in a false-positive event). Monomeric GAF deleted for the POZ multimerization domain also exhibits robust 1D sliding, indicating that multimerization is not required for long-range 1D diffusion.

In living *Drosophila* hemocytes, a population of GAF-FL remains stably bound on chromatin for ~2 min, likely at cognate promoter and enhancer NFRs[26]. Consistent with this finding, our purified GAF-FL remains on free *hsp70* target DNA for a minimum of 40 s. Multimerization is necessary for stable DNA association in vitro, as removal of the POZ domain dramatically reduces GAF dwell time on DNA to ~2 s. The longer GAF-FL dwell time can be explained if its dissociation requires multiple DBDs to simultaneously dissociate from engaged motifs—a low-probability event compared to a single DBD dissociating from one site, consistent with recent modeling predictions[62]. The in vitro specific dwell time of GAF-DBD and GAF-ΔPOZ is similar to the in vivo residence time of the transiently bound GAF-FL population (3.7 s)[26], suggesting that transient binding in vivo can be partially attributed to binding between one DBD (in monomeric or multimeric GAF) and an isolated cognate motif.

Prior evidence has shown that GAF is essential for establishing and maintaining nucleosome-depleted regions in chromatin: (1) GAF is necessary for creating accessible chromatin at inducible, developmental and housekeeping gene promoters[23,29]; (2) it recruits adenosine triphosphate (ATP)-dependent chromatin remodelers to generate accessible chromatin[28,63]; and (3) it exhibits an uncommonly long residence time and high occupancy at target sites in live cells[26]. We find that the target search mode of GAF can be highly influenced by the chromatin environment.

On nucleosome-free DNA, a 1D search can start anywhere on the DNA, whereas nucleosomal targets must be accessed by 3D diffusion and only if sites are rotationally phased to allow 3D binding from solvent. When both free DNA and exposed nucleosomal targets are available, free DNA targets are more readily located by 2–8 fold (Extended Data Fig. 4b), consistent with GAF having a lower $k_{on}$ for nucleosomes

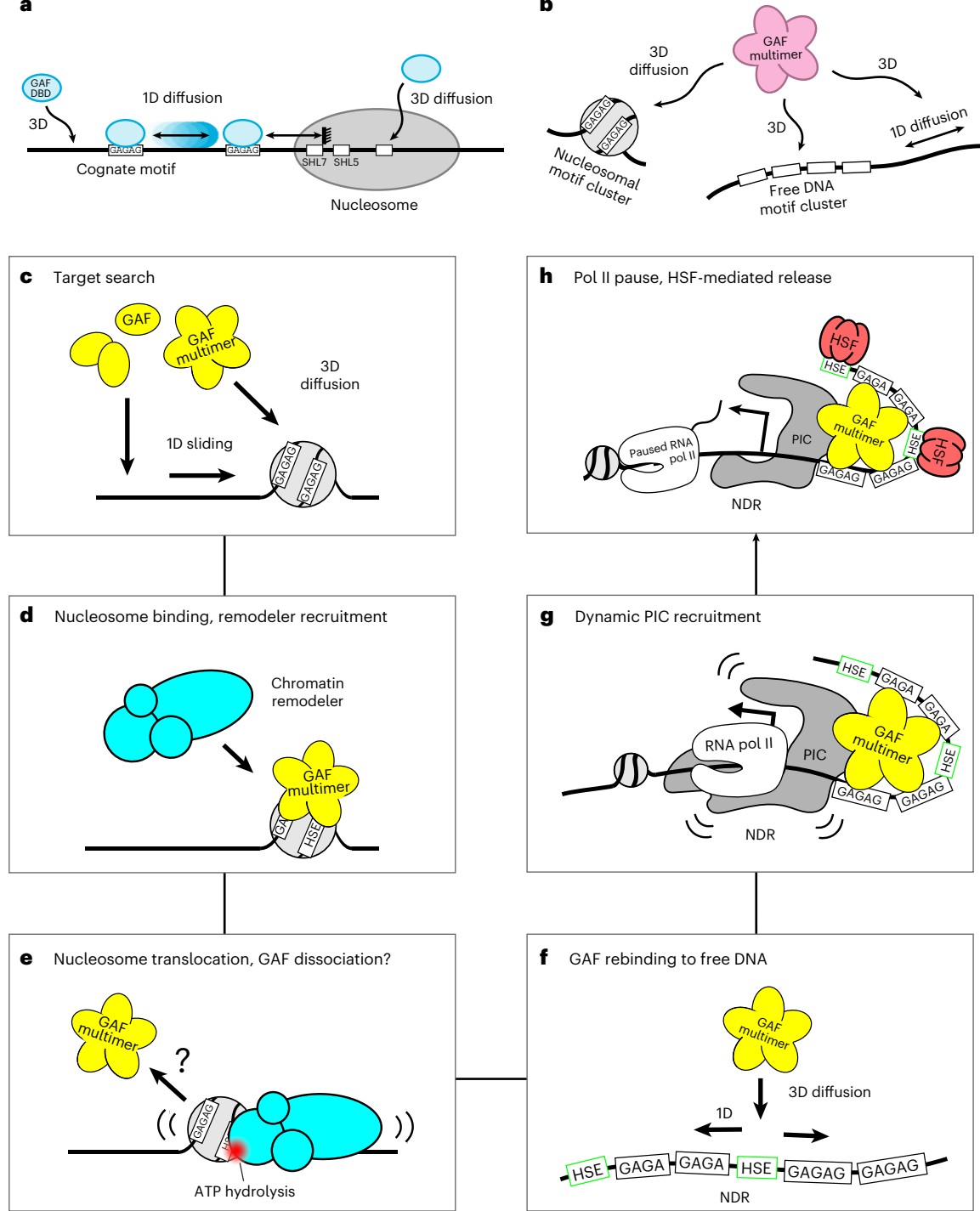

**Fig. 6 | Model for GAF-DBD and GAF-FL target search. a**, GAF-DBD uses two search modes to locate its target on chromatin. In the 1D sliding mode, GAF-DBD lands on an off-target location on free DNA and then slides back and forth to locate the cognate motif (GAGAG). It can escape the cognate site to search for the next site nearby. This 1D search mode allows GAF-DBD to invade into the nucleosome edge but no further. Alternatively, GAF-DBD can also directly associate with a solvent-exposed cognate motif in the nucleosome core from 3D space. This 3D search mode allows GAF to effectively target nucleosomal motifs that are inaccessible by 1D sliding. **b**, GAF-FL uses both 3D and 1D diffusion to locate cognate motif clusters on free DNA. If the motif cluster is inside a nucleosome, GAF-FL can use 3D diffusion for target location.

**c–h**, Hypothetical stepwise model for GAF–remodeler collaboration to mobilize targeted nucleosome for PIC assembly, based on this study and prior literature. **c**, GAF localizes cognate targets on closed chromatin through a combination of 1D sliding and 3D diffusion. **d**, Nucleosome-bound GAF multimer may recruit chromatin remodelers, such as NURF and PBAP. **e**, The recruited ATP-dependent chromatin remodelers shift nucleosomes away from the cognate sites. It is unclear whether GAF dissociates from chromatin during this process. **f–h**, Once the cognate sites become nucleosome free, GAF multimer rapidly locates the free DNA target through 3D and 1D diffusion (**f**), enabling downstream PIC recruitment, transcription initiation (**g**) and binding of other TFs such as HSF (**h**). HSE, heat shock element; NDR, nucleosome-depleted region.

than bare DNA, as do other TFs[11,17]. In contrast, once bound to nucleosomal sites, GAF-DBD dissociation ($k_{off}$) is ~10 times slower from nucleosomal DNA than free DNA, potentially because GAF-DBD binding is stabilized by histones or the DNA superhelix. Our observation of less frequent but longer-lasting binding to nucleosomal cognate sites is consistent with the dissociation rate compensation mechanism reported for the yeast TFs Reb1 and Cbf1 (ref. 17).

In vivo, DNA accessibility is dynamic[64–66], whereby a particular cognate site can alternate between nucleosome-bound and nucleosome-free states because of the opposing activities of ATP-driven chromatin remodelers ('push' and 'pull' from the standpoint of nucleosome-free DNA)[56,65]. In fruit flies, GAF is expressed from an early embryonic stage where it is required for the early establishment and maintenance of nucleosome-depleted regions[29]. Our study provides the molecular basis by which GAF initially locates cognate sites on closed chromatin (Fig. 6c). Upon target recognition, GAF may recruit chromatin remodelers (NURF or PBAP)[67–69] to shift nucleosomes away from cognate sites (Fig. 6d,e), providing accessibility for subsequent recruitment of the PIC or additional TFs such as heat shock factor (HSF) at heat-shock genes or Pho TF and related factors mediating Polycomb-dependent repression at the corresponding DNA elements[70]. Whether GAF must dissociate from the nucleosomal target before the remodeler can proceed or form a ternary complex to prime enzymatic remodeling remains to be determined. Once the cognate sites become nucleosome free, they will be rapidly located by another GAF multimer, residing on target for minutes[26]. Upon GAF dissociation, another GAF multimer rapidly substitutes, achieving high occupancy over time as shown in a separate study[26], forming a potential roadblock to remodeler-driven nucleosome incursions[71,72] (Fig. 6f). The maintenance of promoter accessibility is, thus, achieved by persistent GAF occupancy derived from GAF on–off kinetics, enabling downstream PIC recruitment, pausing of RNA Pol II and binding of additional TFs such as heat-shock-activated HSF (Fig. 6g,h)[73–78].

Lastly, blockage to 1D sliding and the 3D diffusion requirement for robust TF targeting to exposed nucleosome sites has, to our knowledge, not been demonstrated. Because the bulk of genomic DNA (70–90%) is shielded from 1D search by the nucleosome architecture, as shown for other chromatin factors[79], some TFs may exhibit preferential usage of 1D search to bias their search toward NFRs, while others use 3D search to target chromatinized DNA. The interplay involving different TF families, chromatin remodelers and mobilized nucleosomes on dynamically regulated chromatin provides exciting opportunities for future investigations.

## Online content

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

## Methods

### GAF-DBD protein expression and purification

GAF-DBD (Supplementary Information) was cloned into pET SUMO plasmid (Invitrogen, K300-01) using Gibson assembly (New England Biolabs (NEB), E5520S) to contain an additional N-terminal cysteine for fluorophore labeling. The pET 6×His–SUMO–Cys–GAF-DBD plasmid was transformed into *Escherichia coli* (Novagen Rosetta 2 DE3; Sigma-Aldrich, 71400) for protein expression. On the first day, a starter culture was grown from a single colony in 30 ml of terrific broth (Sigma-Aldrich, T9179) and incubated overnight on a 37 °C shaker. On the second day, the starter culture was added to 1 L of terrific broth and incubated at 37 °C until the optical density (OD) reached 0.8. IPTG was added to 0.4 mM final concentration to induce protein expression. Cell pellets were harvested at 5 h after IPTG induction, resuspended in 30 ml of lysis buffer (50 mM Tris pH 7.4, 300 mM NaCl, 10% glycerol and 0.05% Triton X-100) and stored at −80 °C.

The 6×His–SUMO–Cys–GAF-DBD protein was first purified on Ni-NTA agarose (Qiagen, 30210). Briefly, Triton X-100 (0.05% final concentration), protease inhibitor (Roche, 4693132001) and β-mercaptoethanol (BME; 2 mM final) were added to the thawed cell suspension. Cells were lysed by sonication (amplitude level 50, 10 s on, 50 s off, 3-min processing time) on ice. Lysate was clarified by centrifugation at 18,000*g* for 45 min at 4 °C. The supernatant was transferred to 1.25 ml of pre-equilibrated Ni-NTA agarose (2 ml of slurry) in a 15-m; tube and incubated at 4 °C for 1 h with gentle shaking. Then, agarose beads were washed with 10 ml of wash buffer 1 (WB1; 50 mM Tris pH 7.4, 500 mM NaCl, 10% glycerol, 0.05% Triton X-100, 2 mM BME and 20 mM imidazole, pH 8), centrifuged down, resuspended in another 10 ml of WB1 and transferred to a prechilled 2.5-cm gravity flow column. Beads were washed three times with 5 ml of WB1, followed by two times with 5 ml of WB2 (50 mM Tris pH 7.4, 300 mM NaCl, 10% glycerol and 0.05% Triton X-100). To elute the protein, 2 ml of elution buffer (WB2 with 250 mM imidazole) was added to the beads and incubated for 10 min. Then, 1-ml elution fractions were collected immediately after each addition of elution buffer and further 1-ml elution fractions were collected after 5 min of incubation. Typically, ten fractions were collected and most of the protein eluted in the first 2–3 fractions. Peak fractions were pooled and then further purified by cation-exchange chromatography (Cytiva, HiTrap SP HP 17115101) on a fast protein liquid chromatography (FPLC) instrument (Cytiva, ÄKTA).

### Labeling GAF-DBD with Cy3 fluorophore

We followed the 'one-pot' reaction protocol by Jiang et al.[41] to cleave off the 6×His–SUMO using SUMO protease (Invitrogen, K300-01) and label the N-terminal cysteine with Cy3 in a site-specific manner. Briefly, 6×His–SUMO–Cys–GAF-DBD was buffer-exchanged into labeling buffer (100 mM HEPES pH 6.9, 0.5 mM TCEP and 300 mM NaCl) using 24-h dialysis. In parallel, the transesterification reaction of Cy3–NHS (Cytiva, PA13105) was carried out by incubating 0.2 mg of Cy3–NHS in 50 μl of 100 mM HEPES pH 6.9, 0.5 mM TCEP and 500 mM sodium 2-mercaptoethanesulfonate (MESNa; Sigma-Aldrich, PHR1570) for 6 h at room temperature to yield Cy3–MESNa. The one-pot labeling and protease cleavage reaction consisted of 100 μl of dialyzed protein, 10 μl of SUMO protease and 12 μl of 6 mM Cy3–MESNa. The reaction proceeded for 36 h at room temperature. The cleaved 6×His–SUMO and 6×His-tagged SUMO protease were removed by Ni-NTA agarose. The labeling efficiency of GAF-DBD was measured to be 92%.

### Insect cell expression and purification of GAF-FL

Expression of FL-GAF in Sf9 cells was performed using a baculovirus expression construct containing the complete GAF coding sequence with an N-terminal HA tag[37]. Cells harvested from 250 ml of Sf9 cell culture were washed twice in ice-cold PBS. For lysis, 150 ml of ice-cold lysis buffer (1× PBS, 2 mM MgCl₂ 0.1% Triton X-100, 10% glycerol, 1 mM DTT and 1× protease inhibitor cocktail containing 0.17 μg ml⁻¹ PMSF,

0.33 μg ml⁻¹ benzamidine hydrochloride, 1.37 ng ml⁻¹ pepstatin A, 0.284 ng ml⁻¹ leupeptin and 2 ng ml⁻¹ chymostatin) was added to the cell pellet. All further steps were performed at 4 °C. Cells were resuspended and incubated for 5 min. The nuclei were pelleted by centrifugation at 1,500*g* for 4 min. The nuclear pellet was washed once in lysis buffer and salt-extracted in 8 ml of lysis buffer supplemented with 300 mM KCl for 1 h. The soluble proteins were collected by centrifugation at 20,000*g* for 20 min. The nuclear extract was used for anti-HA affinity purification.

A total of 300 μl of 50% anti-HA agarose beads (Sigma, A2095) was added to 8 ml of nuclear extract and incubated for 2.5 h at 4 °C with overhead rotation. The beads were washed five times with a total of 10 ml of ice-cold lysis buffer. The protein bound to the beads was eluted with 100 μl of elution buffer (lysis buffer supplemented with 1 mg ml⁻¹ HA peptide and additional 850 mM NaCl) for 1 h at room temperature with rotation. The final concentration was quantified on a PAGE gel with a BSA standard curve.

### EMSA of FL-GAF and nucleosome core particle

FL-GAF purified from *Drosophila* SF9 cells was mixed and intubated with 3 nM Cy5-labeled nucleosome core particle (3-N-6-Cy5) reconstituted on *hsp70* promoter DNA or Cy5-labeled free DNA (Supplementary Information) at various concentrations in GAF-binding buffer (12.5 mM HEPES–KOH pH 7.6, 0.05 mM EDTA, 6.25 mM MgCl₂, 5% glycerol, 50 mM NaCl, 50 μg ml⁻¹ BSA (Roche, 10711454001) and 0.05% NP-40) for 1 hr at room temperature. The samples were added to 10% sucrose and loaded onto a 1.3% agarose gel in 0.2× TB (18 mM Tris and 18 mM boric acid) pH 8.0 and run for 45 min at 120 V. The gel was imaged for Cy5 fluorescence on the Amersham Typhoon 5 gel imager.

### *E. coli* expression and purification of SNAP-tagged FL-GAF

Maltose-binding protein (MBP)–SNAP–GAF–6×His (SNAP–GAF) was expressed in Lemo21(DE3) competent *E. coli* (NEB, C2528J). Briefly, the expression plasmid was transformed into Lemo21 cells and a single colony was grown overnight in a 30-ml starter culture and then transferred to 1 L of terrific broth. Once the OD reached 0.8, expression was induced by adding IPTG to 400 μM final concentration. The cell pellet was harvested after a 5-h induction at 37 °C, resuspended in 30 ml of lysis buffer (50 mM Tris pH 7.4, 300 mM NaCl, 10% glycerol and 0.05% Triton X-100) and stored at −80 °C.

SNAP–GAF protein was purified on Ni-NTA agarose (Qiagen, 30210), followed by amylose resin (NEB, E8021S) and heparin affinity column (Cytiva, 17040601). Because Ni-NTA selects for the C-terminal 6×His tag and amylose selects for the N-terminal MBP tag, this dual-pulldown strategy enriches full-length proteins with intact N and C termini. Briefly, Triton X-100 (0.05% final concentration), 1× protease inhibitor (Roche, 4693132001) and BME (2 mM final) were added to the thawed cell suspension. Cells were lysed by sonication (amplitude level 50, 10 s on, 50 s off, 3-min processing time) on ice. Lysate was clarified by centrifugation at 18,000*g* for 45 min at 4 °C. The supernatant was transferred to 1.25 ml of pre-equilibrated Ni-NTA agarose (2-ml slurry) in a 15-ml tube and incubated at 4 °C for 1 h with gentle shaking. Then, agarose beads were washed with 10 ml of WB1 (50 mM Tris pH 7.4, 500 mM NaCl, 10% glycerol, 0.05% Triton X-100, 2 mM BME and 20 mM imidazole, pH 8), centrifuged down, resuspended in another 10 ml of WB1 and transferred to a prechilled 2.5-cm gravity flow column. Beads were washed three times with 5 ml of WB1, followed by two times with 5 ml of WB2 (50 mM Tris pH 7.4, 300 mM NaCl, 10% glycerol and 0.05% Triton X-100). To elute the protein, 2 ml of elution buffer (WB2 with 250 mM imidazole) was added to the beads and incubated for 10 min. Then, 1-ml elution fractions were collected immediately after each addition of elution buffer and further 1-ml elution fractions were collected after 5 min of incubation. Typically, ten fractions were collected and most of the protein eluted in the first 2–3 fractions.

To pull down the MBP-tagged SNAP–GAF, peak fractions from the Ni-NTA purification were combined and mixed with pre-equilibrated amylose resin and then incubated at 4 °C for 1 h with gentle shaking. Amylose resin was then washed with 10 ml of WB1 twice and transferred to a prechilled 2.5-cm gravity flow column. The resin was washed three times with 5 ml of WB1, followed by two times with 5 ml of WB2. To elute the protein, 2 ml of MBP elution buffer (WB2 + 10 mM maltose) was added before incubating for 10 min. Elution fractions were collected, 1 ml at a time, similar to the Ni-NTA elution step. Peak fractions were pooled and further purified on a heparin column using FPLC.

## Histone purification and octamer refolding

*Drosophila* histones H2A, H2B, H3 and H4 were separately expressed in bacteria and purified as previously described[80]. Briefly, BL21(DE3) pLysS or Rosetta(star) cells were induced with 0.3 mM IPTG for 3 h at 37 °C, harvested and frozen in WB (50 mM Tris-HCl pH 7.5, 100 mM NaCl and 1 mM EDTA, pH 8.0). For each histone, cells were then lysed with sonication and histone protein was purified as inclusion bodies, with several cycles of resolubilizing the pellet followed by centrifugation. After the last spin, the pellet was stored at −20 °C, smeared on the inside of a 50-ml conical tube. These inclusion bodies were then solubilized with 1 ml of DMSO followed by 40 ml of unfolding buffer (7 M guanidine-HCl, 20 mM Tris-HCl pH 7.5 and 10 mM DTT) with agitation. After exchanging the sample into 7 M urea, 10 mM Tris-HCl pH 7.5, 100 mM NaCl and 1 mM EDTA (pH 8.0), the sample was further purified over tandem Q-SP ion-exchange columns, with purified protein eluted from the SP column with a NaCl gradient. Purified protein was dialyzed against water plus 5 mM BME, lyophilized in 2-mg aliquots and stored at −20 °C.

Histone octamer was produced as previously described[81]. Briefly, equal amounts of each histone (typically 4 mg) were solubilized in unfolding buffer (6 M guanidinium chloride, 20 mM Tris-HCl pH 7.5 and 5 mM DTT) for 30 min at room temperature and then all histones were combined in an equimolar ratio and dialyzed against refolding buffer (2 M NaCl, 10 mM Tris-HCl pH 7.5, 1 mM EDTA and 5 mM BME) at 4 °C overnight, with three dialysis changes. The pooled histones were then concentrated and the histone octamer purified by size-exclusion chromatography. Purified octamer was concentrated, aliquoted and stored at −80 °C.

## Two-color TIRF microscope

The method for two-color smFRET microscopy was previously described[50].

## Two-color smFRET imaging of Cy3–GAF-DBD binding to Cy5-labeled DNA

Two complementary single-stranded DNA molecules, one containing an internal Cy5 fluorophore and another containing a 5′ biotin, were annealed to form Cy5-labeled DNA constructs (Supplementary Information). On a homemade PEG-passivated and sparsely biotinylated flow channel[82,83], the following reagents were flowed in order: 100 µl of T50 buffer (10 mM Tris-HCl pH 8 and 50 mM NaCl), 40 µl of 0.2 mg ml⁻¹ Neutravidin in T50 (1-min incubation; Thermo Fisher, 31000), 100 µl of T50, 50 µl of 12.5 pM Cy5–DNA (3-min incubation), 100 µl of T50, 50 µl of 0.2 nM Cy3–GAF-DBD in imaging buffer (50 mM NaCl, 50 µg ml⁻¹ BSA (Roche, 10711454001), 0.05% NP-40 (Sigma, I8896), 12.5 mM HEPES–KOH pH 7.6, 0.05 mM EDTA, 6.25 mM MgCl₂, 5% glycerol, 0.8% w/v dextrose, 2 mM Trolox, 1 mg ml⁻¹ glucose oxidase (Sigma-Aldrich, G2133) and 500 U per ml catalase (Sigma-Aldrich, C3155)). Note that Cy3–GAF-DBD was initially diluted to 20 nM using storage buffer (50 mM Tris pH 7.4, 300 mM NaCl, 10% glycerol and 0.05% Triton X-100) to prevent aggregation. The channel was imaged under a two-color total internal reflection fluorescence (TIRF) microscope with a 10-Hz frame rate. Laser excitation was programmed to be 10 frames of Cy5 excitation followed by 990 frames of Cy3 excitation for a 100-s video.

## Two-color smFRET data analysis

Two-color videos were converted to single-molecule fluorescence intensity time trajectories using custom-written IDL scripts and analyzed using custom-written MATLAB scripts. Dwell time data were manually collected by recording the start and end time of each event. More than 100 dwell times were collected per technical replicate. Then, $1 - \mathrm{CDF}$ (cumulative distribution function) plots of the dwell times were fit to a single-component exponential decay function (ExpDec1 function: $y = y_0 + e^{-x/\tau}$) in OriginPro. Errors were calculated as the s.e.m. of $\tau$ values from fitting each replicate. Binding frequency data were calculated by manually counting the total number ($n > 100$ for one replicate) of binding events and dividing the total number of events by the product of video length and number of trajectories examined. Binding frequencies from multiple replicates were averaged to obtain the final binding frequency.

## Labeling DNA oligo with Cy5 or Cy7

Single-stranded DNA oligos were site-specifically labeled with Cy5–NHS (Cytiva, PA15100) or Cy7–NHS (Lumiprobe, 25320) through an amino group attached to an internal thymine through a six-carbon linker (Integrated DNA Technologies, iAmMC6T). A 62.5-µl labeling reaction contained 160 µM amino-modified oligo, 200 mM freshly dissolved NaHCO₃, 8 mM NHS dye and nuclease-free water. The reaction was incubated with gentle mixing for 4 h at room temperature and then overnight at 4 °C. The labeled oligo was purified by ethanol precipitation to remove excess dye. Cy7-labeled oligo typically requires two rounds of ethanol precipitation. If the labeling efficiency was lower than 70%, a second round of labeling reaction was repeated on the labeled oligo. The final labeling efficiencies were typically 80–90%.

## Cy5–Cy7 dual-labeled DNA construction

Making a DNA construct dual-labeled at our desired positions was an engineering challenge because a one-step PCR reaction would require an internally labeled DNA oligo to be longer than 100 bp. Such DNA oligos were not produced by IDT at the time (now available by requesting an ultramer). To overcome this challenge, a shorter dual-labeled DNA was made first by PCR, restriction-digested to produce a sticky overhang and then ligated with a biotinylated DNA fragment to form the complete construct. Briefly, a 2.4-ml PCR reaction was performed using GoTaq buffer (Promega, M7921), 200 µM dNTPs, 1 µM forward primer, 1 µM reverse primer, 100 ng or less template DNA and 24 µl of Taq DNA polymerase (NEB, M0273L) (primer sequences in Supplementary Information). The PCR DNA product was purified and concentrated by ethanol precipitation. The DNA was digested with DraIII-HF restriction enzyme (NEB, R3510L) for 3 h at 37 °C. The digested DNA product was purified by anion-exchange chromatography (Cytiva, 17115301) on an FPLC instrument, followed by ethanol precipitation of DNA in the peak fractions. The biotinylated DNA fragment was simultaneously prepared by annealing two single-stranded DNA oligos. The Cy5–Cy7 dual-labeled DNA was ligated with the biotinylated fragment using T4 DNA ligase (Thermo Fisher, EL0011) and the ligated DNA was purified by agarose gel extraction.

## Cy5–Cy7 dual-labeled 601 nucleosome reconstitution

Nucleosome reconstitution was carried out using salt gradient dialysis as detailed previously[81]. Briefly, nucleosomes were reconstituted by combining *Drosophila* octamer and nucleosomal DNA together, each at a concentration of approximately 262 nM, in buffer composed of 10 mM Tris pH 7.5, 2 M NaCl, 1 mM EDTA, 1 mM BME and 0.1 mg ml⁻¹ BSA. For 601 DNA constructs with GAF-binding sites, 6.4 g of Lambda DNA (New England Biolabs) was also added to prevent any aggregation that could occur in an excess of histone proteins. When reconstituting the *hsp70* promoter nucleosomes, Lambda DNA was excluded as it would result in lower yield. The nucleosome sample was placed in a 7-kDa-MWCO (molecular weight cutoff) dialysis button (Slide-A-Lyzer MINI dialysis unit, Thermo Fisher Scientific) and dialyzed from 600 ml of high-salt

buffer (10 mM Tris pH 7.5, 2 M NaCl, 1 mM EDTA, 3.5 mM BME and 0.02% NP-40) into low-salt buffer (10 mM Tris pH 7.5, 50 mM NaCl, 1 mM EDTA, 3.5 mM BME and 0.02% NP-40) overnight using gradient dialysis. After dialysis, nucleosomes were concentrated to a volume of 100 L and purified over a 4-ml sucrose gradient (5–20%). After purification, the nucleosome-containing fractions were identified by agarose gel electrophoresis, pooled, concentrated using a 10-kDa-MWCO centrifugal concentrator and dialyzed into a low-salt nucleosome reconstitution buffer overnight. The quality of nucleosomes was assayed using 6% native PAGE. Nucleosomes were also quantified by SYBR green (Thermo Fisher Scientific) staining. The intensity of the nucleosome band was compared to the intensity of a known quantity of DNA standards on the same 6% native PAGE. Samples were stored in the dark at 4 °C and used for approximately 1 month.

### Cy5–Cy7 dual-labeled hsp70p nucleosome reconstitution

The DNA sequence of the *hsp70p* nucleosome was a modified 187-bp fragment of the *D. melanogaster hsp70* promoter. According to previously mapped nucleosome positions on a longer *hsp70* promoter fragment, our 187-bp DNA should form a 40-N-0 nucleosome, where the 147-bp nucleosomal DNA (N) contains all nine near-cognate and cognate motifs (GAG, GAGA, GAGAG, etc.) on the 187-bp DNA. To mutate seven of the motifs, leaving only two GAF motifs intact, we swapped the seven motifs with the corresponding nucleotides on the Widom 601 sequence[46] to preserve the nucleosome-forming properties of the DNA. To ensure that the nucleosome formed at the desired position on DNA, we followed the nucleosome ligation protocol in Huh et al.[48]. Briefly, we reconstituted the NCP using a 150-bp + 3-nt DNA, heat-shifted the NCP at 55 °C for 30 min and then used T4 ligase to attach the 37-bp + 3-nt biotinylated linker DNA by incubating at 4 °C for 12 h.

### Three-color TIRF microscope

The methods for three-Color smFRET microscopy was previously described[50].

### Three-color imaging of Cy3–GAF-DBD binding to Cy5–Cy7-labeled DNA or nucleosome

On a homemade PEG-passivated and sparsely biotinylated flow channel[82,83], the following reagents were flowed in order: 100 µl of T50 buffer, 40 µl of 0.2 mg ml⁻¹ Neutravidin in T50 (1-min incubation), 100 µl of T50, 50 µl of 50 pM Cy5–Cy7-labeled nucleosome or DNA (3-min incubation), 100 µl of T50 and 50 µl of 0.1 nM (unless stated otherwise) Cy3–GAF-DBD in imaging buffer. GAF-DBD concentrations were optimized to maximize the number of single GAF-DBD-binding events per trace while minimizing the number of more than two GAF-DBD molecules bound at the same time. The channel was imaged under a three-color TIRF microscope with a 28.6-Hz frame rate (35-ms exposure time per frame) for 601 nucleosomes with GAF motifs (Fig. 3) or a 10-Hz frame rate for *hsp70p* nucleosome (Fig. 4). Laser excitation was programmed to be 10 frames Cy7, 10 frames Cy5, 960 frames Cy3, 10 frames Cy7 and 10 frames Cy5.

### Three-color data analysis (dwell time collection, 1 − CDF fitting, converting traces to colored stripes)

Three-color videos were converted to single-molecule fluorescence intensity time trajectories using custom-written IDL scripts and analyzed using custom-written MATLAB scripts. Binding event dwell times were manually collected by recording the start and end time of each event. Only single GAF-DBD-binding events were analyzed. Single-molecule trajectories were converted into rastergrams using custom-written MATLAB scripts for visualization and motif dwell time collection, described below. The 1 − CDF plot of binding event dwell time or motif dwell time histogram was fit to the ExpDec1 function in Origin. For Fig. 3, nucleosomes containing with both Cy5 and Cy7 fluorophores were selected for analysis. Binding events were manually

categorized into nucleosome and linker DNA binding (FRET alternates between Cy5 and Cy7 in the same binding event), linker DNA binding only (Cy3–Cy5 FRET only) or nucleosome binding only (Cy3–Cy7 FRET only) (Fig. 3h). The landing site of GAF-DBD on SHL7-601 (Fig. 3i) nucleosome was analyzed by manually categorizing the initial FRET pattern of binding events into (1) landing on linker DNA and sliding to nucleosome (transition from Cy3–Cy5 to Cy3–Cy7 FRET); (2) landing on nucleosome and sliding to linker DNA (transition from Cy3–Cy7 to Cy3–Cy5 FRET); (3) linker DNA binding only (Cy3–Cy5 FRET throughout binding); (4) nucleosome binding only (Cy3–Cy7 FRET throughout binding); (5) nonspecific binding and sliding to linker DNA (Cy3 signal only to Cy3–Cy5 FRET); and (6) nonspecific binding and sliding to nucleosome (Cy3 signal only to Cy3–Cy7 FRET).

For Extended Data Fig. 4, nucleosomes with Cy7 fluorophore were selected for analysis. Binding events were categorized into (1) linker motif binding only if it shows constant Cy3–Cy5 FRET and no Cy3–Cy7 FRET throughout; (2) nucleosomal motif binding only if it shows constant Cy3–Cy7 FRET and no Cy3–Cy5 FRET throughout; (3) scanning if it exhibits dynamic transitions between Cy3–Cy7 FRET and either Cy3-only or Cy3–Cy5 FRET during binding; or (4) other if none of the above applies.

### Binding site assignment for three-color FRET data

Three-color FRET trajectories with clear alternating Cy5–Cy7 emission (Figs. 2 and 3 and Extended Data Fig. 3b,h) were fit to an HMM (Supplementary Note) for binding site assignments. The remaining trajectories (Fig. 3 and Extended Data Fig. 3e) were analyzed with a custom-written relative intensity algorithm. Briefly, the algorithm assigns binding at a specific time point to the Cy5 site if Cy5 fluorescence is stronger than Cy7 or to the Cy7 site if Cy7 fluorescence is stronger. If it detects only Cy3 fluorescence, then nonspecific binding is assigned.

### Biotinylated concatenated plasmid DNA preparation

The *Drosophila hsp70* promoter sequence was inserted into the 3.32 pBluescript cloning vector. First, 25 µg of the plasmid containing the *hsp70* promoter sequence was digested with HindIII restriction enzyme (NEB, R3104) for 3 h at 37 °C. The digested fragment was then purified using a Qiagen DNA purification kit (Qiagen, 20021). Next, 6 µg of the purified fragments were concatenated and biotinylated using the Adaptor 1 Lumicks DNA repeat assembly kit designed for HindIII overhangs (Lumicks, 00029). The same was performed for the empty vector control that did not contain the *hsp70* promoter sequence or any other GAGAG motifs. Success of the assembly reaction was confirmed by gel shift on agarose gel (Extended Data Fig. 7a).

### Optical tweezers and confocal microscopy data collection

Optical tweezers experiments were performed at room temperature on a Lumicks C-Trap configured with two optical traps. Imaging was performed using a Lumicks five-channel microfluidic flow cell as previously described[51,56]. Because of the heterogeneity in concatenated plasmid length, DNA tethered between the beads was first checked by a force–distance (FD) curve to ensure a proper double-stranded DNA and identify the distance between the beads. FD curves were collected by bringing the beads 5 µm apart, zeroing the force, extending to maximum stretch at 60 pN and exporting as FD curves from the Bluelake software and plotted using Lumicks Pylake (Extended Data Fig. 7b). Distance between the beads at a force of 5 pN was used for determining the length of the DNA tether and number of *hsp70* promoters present (Extended Data Fig. 7b). Traps were then moved to the protein channel and protein was flown on the tether stretched to 5 pN at 0.3 bar. AF488–GAF-FL or AF546–GAF-ΔPOZ was flown through the protein channel diluted to 80 nM in imaging buffer (50 mM NaCl, 50 g ml⁻¹ BSA (Roche, 10711454001), 0.05% NP-40 (Sigma, I8896), 12.5 mM HEPES–KOH pH 7.6, 0.5 mM EDTA, 6.25 mM MgCl₂, 0.8% w/v dextrose and 2 mM Trolox). Flow was turned off before imaging. Kymographs were generated by

setting a confocal scan line between the beads at a pixel time of 0.1 ms and size of 100 nm. Because of the differences in the DNA tether length from heterogenous concatenation, the confocal scan line was modified before imaging each new tether to make sure it spans the entire length of the DNA. This resulted in a frame rate range of 70–200 ms.

## Single-particle tracking and diffusion analysis

Raw kymograph files were processed using the Lumicks custom software Pylake version 1.0.0 python package. All scripts for kymotracking and diffusion analysis are detailed in the Lumicks Pylake user guide (https://lumicks-pylake.readthedocs.io/en/v1.0.0/theory/diffusion/diffusion.html). Molecules were tracked using the kymotrack widget with centroid refinement and track width set to 0.5. Output of each tracked molecule gave position, time and summed photon count that were used for downstream analysis. MSD values were obtained for each trace using the Lumicks msd function with maximum lags set to 5. Because molecules exhibited the same diffusion pattern in each condition, averages of the MSD values for each time lag from each trace were reported (Fig. 5g). Diffusion constant was measured using the Lumicks estimate_diffusion function for ordinary least squares linear regression of the MSD values. This was performed for each trace and the diffusion constant was averaged for each condition. Statistical analysis was performed using an unpaired Student's $t$-test in Prism; $P$ values are reported in figure legends.

## Theoretical estimation of short-range 1D diffusion preceding apparent direct binding

If we assume no direct 3D binding when GAF-FL searches for its cognate site on the positive-strand $hsp70$ DNA, the average distance it needs to slide before reaching the closest site would be one plasmid length (1 μm, 3.3 kb). Furthermore, 1D diffusion across 1 μm within our confocal exposure time (0.2 s) requires a $D_{coef}$ larger than 2.5 μm² s⁻¹. With the measured $D_{coef}$ for GAF-FL on vector-only DNA, 0.149 μm² s⁻¹, GAF-FL would only be able to slide 0.25 μm. Thus, if GAF falls within 0.25 μm of the cognate site and diffuses to the target, we might falsely call this a 3D binding event. The probability of falling onto a 0.25-μm segment on 1 μm is 25%. This is also the probability of falsely calling a short-range 1D diffusion as a 3D event, which is less than 44% (the fraction of binding events observed to be direct binding).

## Mass photometry of GAF-FL and GAF-ΔPOZ

Mass photometry experiments were performed on a Refeyn TwoMP instrument in the same imaging buffer used in the optical tweezers experiment without any detergent (NP-40) and technical steps were followed as previously described[84]. Briefly, a molecular weight standard curve was first generated using a mixture of known protein standards. BSA (66 kDa), alcohol dehydrogenase (147 kDa) and thyroglobulin (670 kDa) were diluted in buffer to 2× concentrations of 1.97 nM, 17.7 nM and 1.32 nM, respectively. Then, 10 μl of buffer was placed in the well to focus the mass photometer. Subsequently, for sample imaging, 10 μl of the 2× protein standard mix was mixed in the buffer-containing well. Three distinct known molecular weight species from the calibrants were used to generate the calibration curve for conversion of contrast counts to mass units. GAF-FL and GAF-ΔPOZ were imaged the same way with 10-min incubation before measuring. Concentrations reported are the final concentration in the imaging well. Results were fit to a Gaussian using the Refeyn DiscoverMP software.

## Reporting summary

Further information on research design is available in the Nature Portfolio Reporting Summary linked to this article.

## Data availability

All data needed to evaluate the conclusions are present in the paper and Supplementary Information. Raw data in PMA format for TIRF and H5 format for 2D confocal scans and kymographs are available upon request. Raw single-molecule trajectories shown in the paper are provided through Zenodo (https://doi.org/10.5281/zenodo.15702401)[85]. Source data are provided with this paper.

## Code availability

Code used for TIRF data processing and trace analysis is available from GitHub (https://github.com/ashleefeng/gaf_target_search). Code used for HMM is also available from GitHub (https://github.com/Yibenfu/HMM_3colorFRET).

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

## Acknowledgements

We thank all members of the T.H. and C.W. laboratories, especially R. Merino Urteaga, J. Hao and T. Liao for support and suggestions and S. Pangeni, P. Meneses and C. Carcamo for their experience with optical tweezers and offer of many technical suggestions that guided our data collection and analysis. We thank P. Verrijzer (Erasmus MC) for the gift of plasmids. We thank Y. Li at the Johns Hopkins Medicine International Eukaryotic Tissue Culture Facility for FL-GAF expression in insect cells. We thank E. Lin and R. He for pilot protein purifications. We thank B. Cesar for mass photometry training and sharing of protein standards. This study was supported by the National Institutes of Health grants U01 DK127432 (T.H., C.W. and M.E.J.), R35 GM149291 (C.W.), R35 GM122569 (T.H.) and S10 OD025221 (T.H.) and the Johns Hopkins Discovery Award (C.W., G.D.B. and T.H.).

## Author contributions

C.W. conceptualized the project. X.A.F., M.Y., T.H. and C.W. designed the experiments. X.A.F., K.M.N. and C.L. generated the DNA constructs, recombinant proteins and nucleoprotein complexes for TIRF measurements. I.A. and G.D.B. purified and reconstituted the histone octamer. X.A.F. and K.M.N. performed the TIRF measurements. X.A.F. wrote the software and performed the data analysis for TIRF data. X.A.F., M.Y. and K.M.N. prepared the DNA and proteins for optical tweezer experiments. M.Y. performed the optical tweezer measurements and data analysis. Y.F., X.A.F and M.E.J. wrote the software for HMM. X.A.F., M.Y., Y.F., M.E.J., G.D.B., T.H. and C.W. wrote the paper. T.H. and C.W. supervised the project.

## Competing interests

The authors declare no competing interests.

## Additional information

**Extended data** is available for this paper at https://doi.org/10.1038/s41594-025-01643-0.

**Correspondence and requests for materials** should be addressed to Taekjip Ha or Carl Wu.

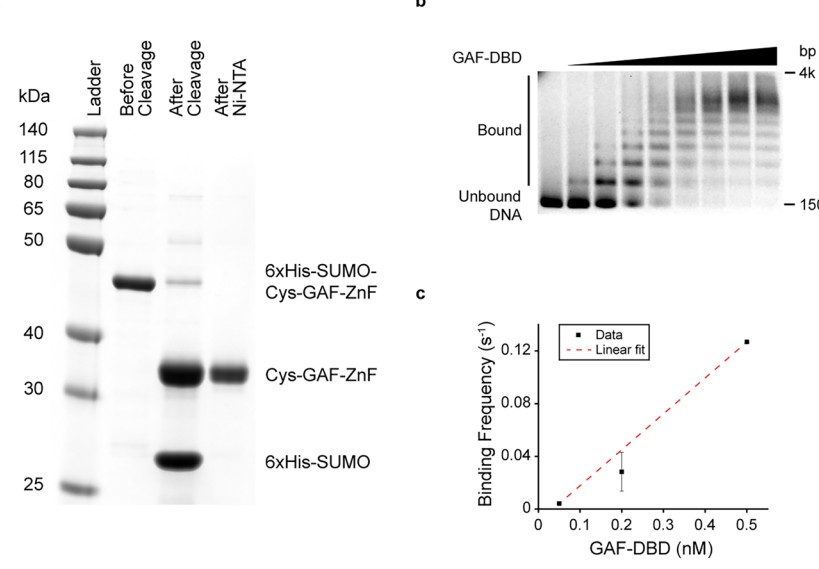

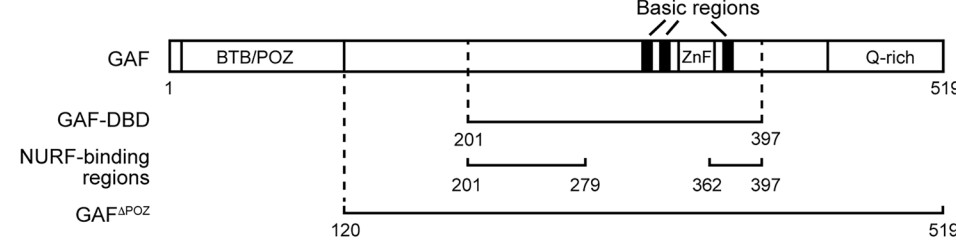

**e**

*hsp70* promoter

```
  1 CAGATCTGAATTGACGCTCCGTCGACGAAGCGCCTCTATTTATACTCCGG
 51 CGCTCTTTTCGCGAACATTCGAGGCGCGCTCTCTCGAAGCAACGAGAACA
101 GTGTGCCGTTTACTGTGCGACAGAGTGAGAGAGCAATAGTACAGAGAGGG
151 AGAGTCACAAAACGAATAGAGAATAACGGCCAGAGAAATTTCTCGAGTTT
201 TCTTTCTGCCAAACAAATGACCTACCGCAACAACCAGTTTGTTTTGGGAT
251 TCTAGAATATTCGCTTTATTTTGGAAATTTCTTTATAAATACGGCTGCTT
301 AAATTAATTATGGTAGAGATAATCGCAGAGCTTGGTTATGCTTATCGTGG
351 ATCCGAATT
```

**Extended Data Fig. 1 | Purification of GAF-DBD protein. a**, Subtractive Ni-NTA purification after "one-pot" reaction which cleaved off 6xHis-SUMO and labeled the N-terminus of GAF-DBD with Cy3. The calculated molecular weights are 34.4 kDa for 6xHis-SUMO-Cys-GAF-DBD, 21.0 kDa for Cys-GAF-DBD and 13.4 kDa for 6xHis-SUMO. This experiment was independently repeated once with similar results. **b**, EMSA showing GAF-DBD binds to hsp70 promoter NCP DNA with $K_{1/2}$ ~ 12 nM. GAF-DBD concentrations are, from left to right, 0, 4, 8, 12, 16, 20, 24, 28, 32 nM and the DNA concentration is 3 nM. This experiment was independently repeated once with similar results. **c**, The binding frequency of GAF-DBD on cognate DNA as a function of protein concentration (For each concentration, N = 40 traces from two imaging sessions were used to quantify the binding frequency). The binding constant is determined from a linear fit to the data ($R^2$ = 0.99) to be 0.27 $s^{-1}$ $nM^{-1}$. To calculate $K_D$, we divide the measured $k_{on}$ (3.1 $s^{-1}$, Fig. 1) by this number to get 11.5 nM which agrees well with the 12 nM apparent $K_D$ ($K_{1/2}$) determined from the EMSA in b. Data are represented as mean values +/− SD. **d**, Schematics of GAF-FL short isoform, GAF-DBD and NURF-binding regions[69] to scale. **e**, Native *hsp70* promoter DNA sequence. GAF cognate sites are highlighted in red. TATA box in bold.

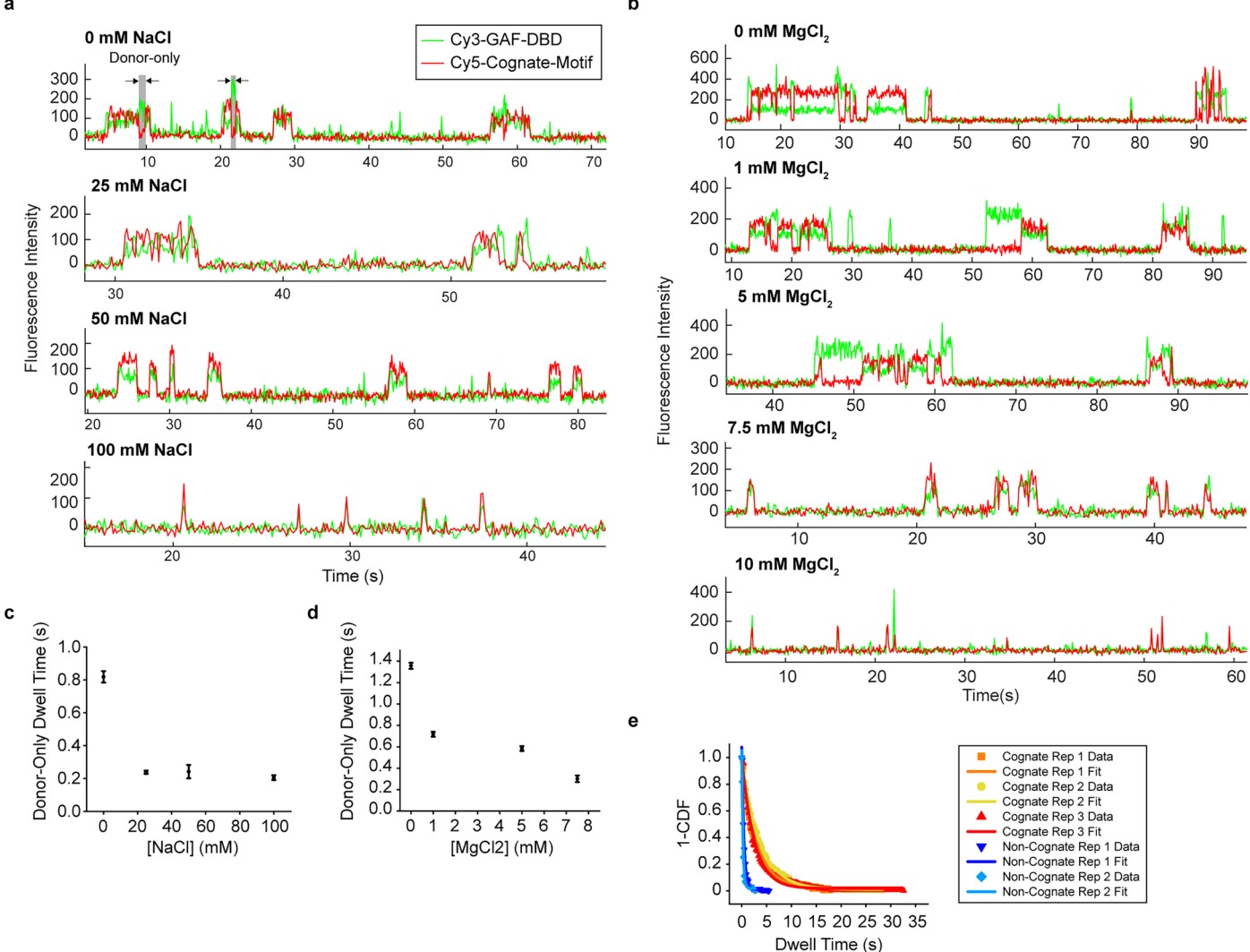

**Extended Data Fig. 2 | Donor-only dwell time is inversely correlated with ionic strength of buffer solution. a**, Representative trajectories showing Cy5-cognate DNA bound by Cy3-GAF-DBD in 0, 25, 50, or 100 mM NaCl. Grey-highlighted durations indicate donor-only dwell times. **b**, Representative trajectories showing Cy5-cognate DNA bound by Cy3-GAF-DBD in 0, 1, 5, 7.5, or 10 mM MgCl$_2$. The sensitivity of FRET dynamics to salt concentration suggests that the transient donor-only periods are not due to photo-induced blinking. **c**, Donor-only dwell time ($\tau$ from fitting 1-CDF to a single-exponential decay; see Methods) as a function of NaCl concentration (N = 119, 147, 109, 23 for 0 mM, 25 mM, 50 mM, and 100 mM, respectively), and **d**, MgCl$_2$ concentration (N = 177, 128, 116, 149 for 0, 1, 5, and 7.5 mM, respectively). Data are represented as mean values +/- standard errors for a single-component exponential decay fit. DNA is the same as cognate DNA in Fig. 1. **e**, 1-CDFs for dwell times reported in Fig. 1h with fits to a single-component exponential decay function (ExpDec1 in OriginPro). Replicates were fit individually and their average $\tau$ is reported as the final dwell time.

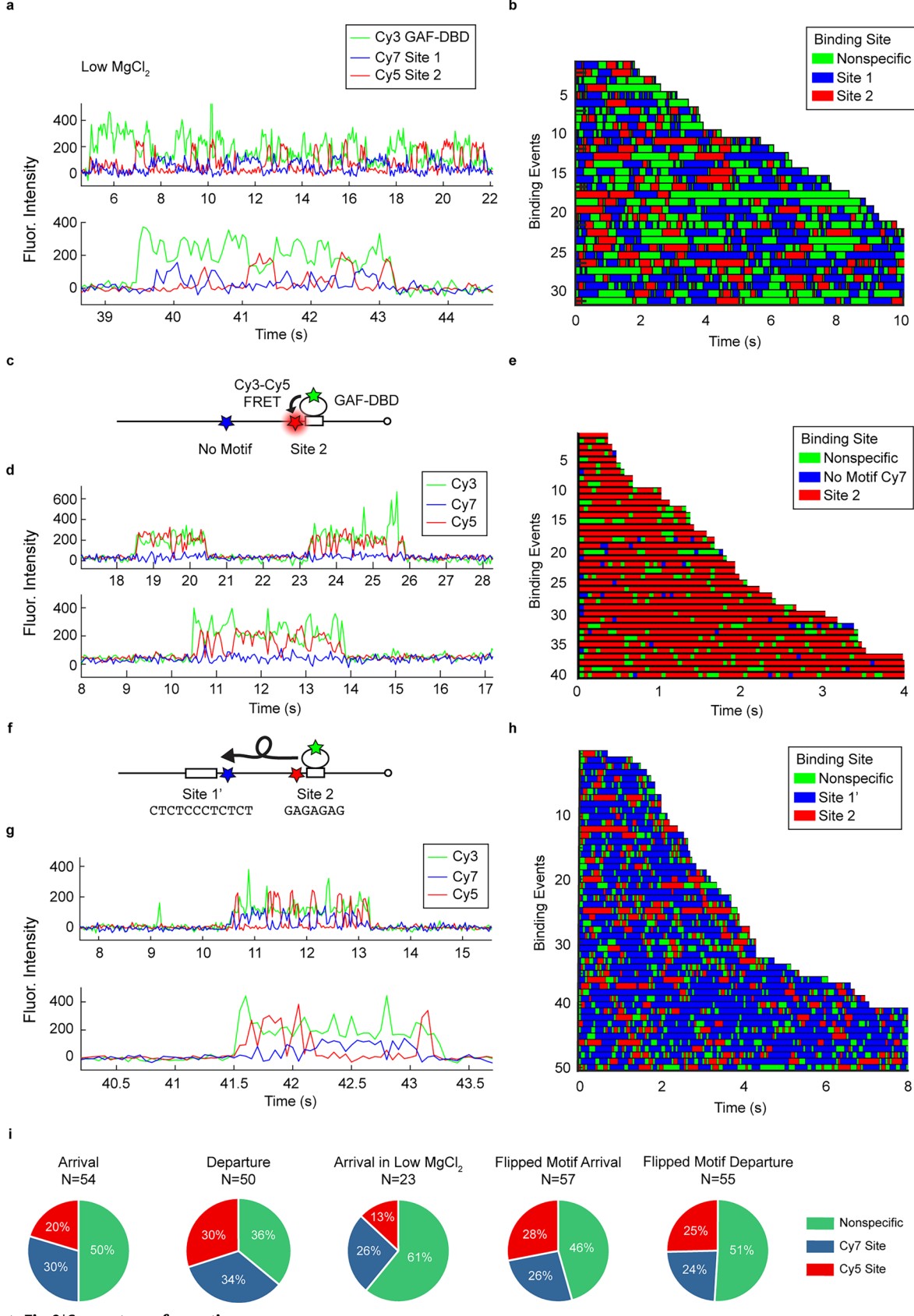

**Extended Data Fig. 3 | See next page for caption.**

**Extended Data Fig. 3 | GAF sliding kinetics on Cy5 & Cy7 dual-labeled DNA as a function of salt concentration, cognate site and motif orientation.**
**a**, Representative single-molecule trajectories of Cy5 & Cy7 DNA bound by Cy3-GAF-DBD in low MgCl$_2$ (3 mM). **b**, Rastergram of 31 Cy5 & Cy7 DNA molecules bound by Cy3-GAF-DBD at 3 mM MgCl$_2$. **c**, Schematic of Site 2 Only construct where Site 1 was replaced with non-cognate sequence. **d**, Representative single-molecule trajectories of Site 2 Only DNA bound by GAF-DBD. **e**, Rastergram of 40

Site 2 Only DNA molecules bound by GAF-DBD. **f**, Schematic of the flipped motif DNA construct where Site 1 is replaced with its complementary sequence, Site 1'. **g**, Representative single-molecule trajectories of flipped motif DNA bound by GAF-DBD. **h**, Rastergram of 57 flipped motif DNA molecules bound by GAF-DBD. DNA same as Fig. 4. **i**, Categories of GAF-DBD arrival landing site and departure launching site.

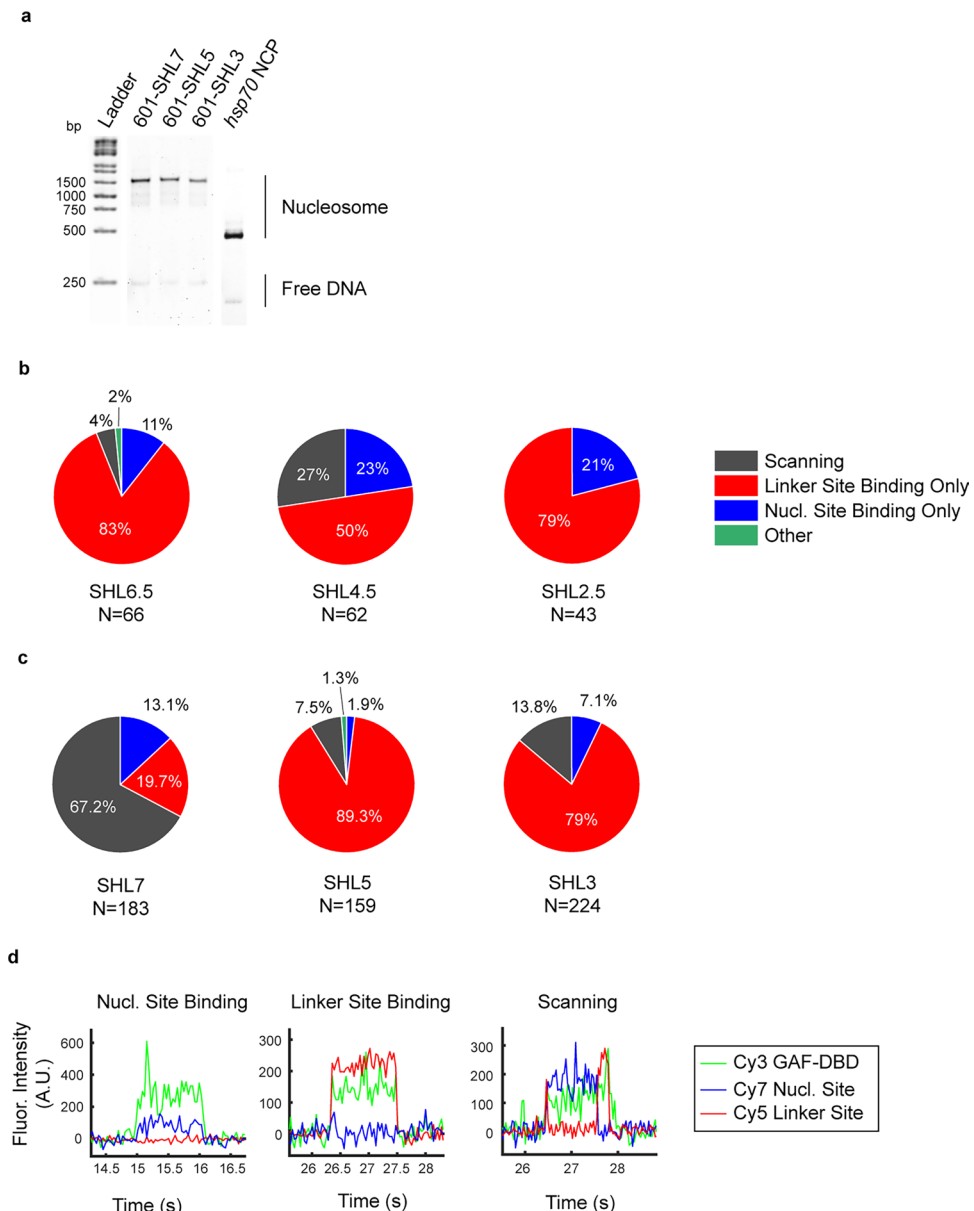

**Extended Data Fig. 4 | Nucleosomal motif accessibility depends on helical phasing. a**, Representative native PAGE gels for nucleosome constructs used in this study. All nucleosome constructs used in this study were assessed by native PAGE gels and showed similar results. **b**, Classification of Cy3-GAF-DBD binding events on '601' nucleosomes (40-N-40) with a cognate site placed at SHL6.5, SHL4.5, and SHL2.5. Classification is based on Cy3-Cy7 FRET dynamics, see details in methods. **c**, Classification of Cy3-GAF-DBD binding events on same '601' (40-N-

40) nucleosomes with cognate site placed at SHL7, SHL5, and SHL3 (same dataset as Fig. 3, re-analyzed for direct comparison with **a**). **d**, Representative single-molecule trajectories for each category of 601-SHL6.5, -SHL4.5, and -SHL2.5 nucleosomes. Some scanning events show prolonged Cy3-Cy7 FRET (right) which appear distinct from scanning events on the 601-SHL7 nucleosomes (Fig. 3b). Note that scanning events (events that show dynamic Cy3-Cy7 FRET) may also come from free DNA which contaminates the nucleosome sample.

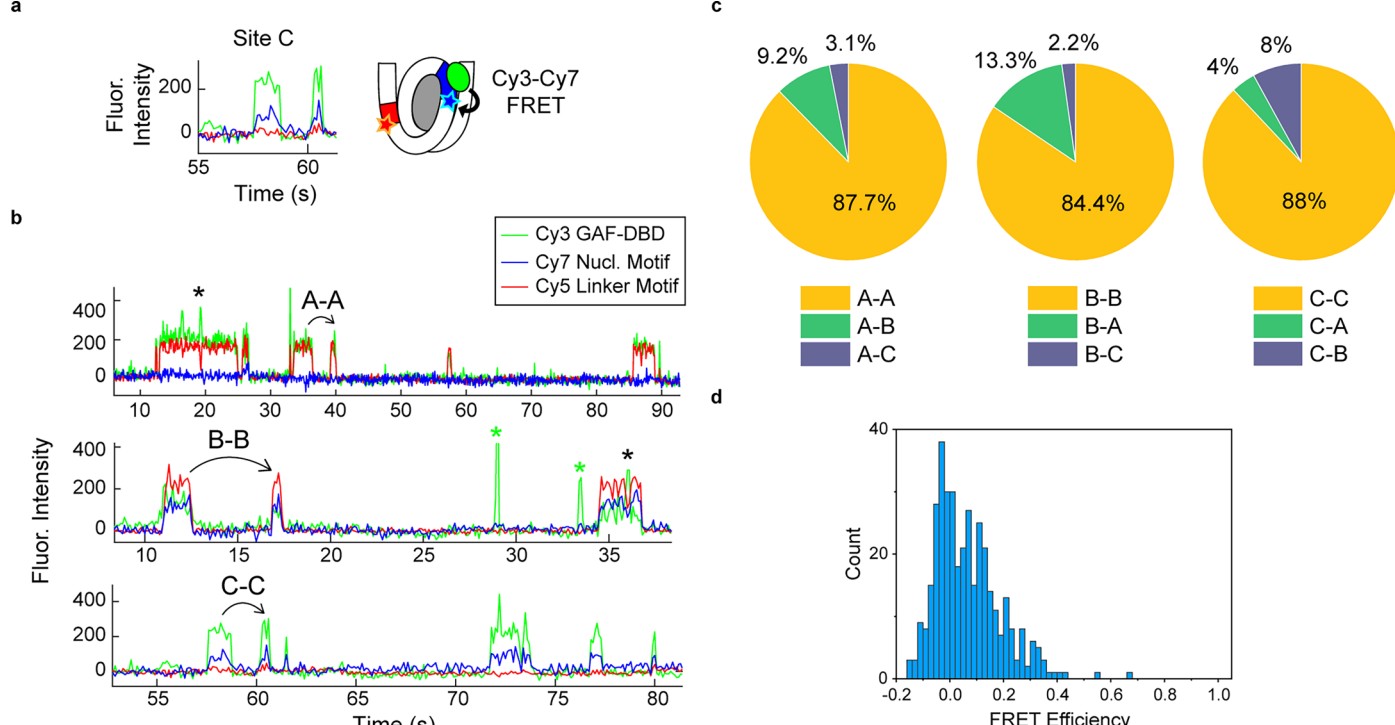

**Extended Data Fig. 5 | GAF-DBD preferentially re-visits the same cognate site on individual hsp70 nucleosomes. a**, Representative single-molecule trajectory for Site C binding. **b**, Representative single-molecule trajectories showing repetitive visits to the same binding site on a single nucleosome. Upper trace shows repetitive visits to Site A; middle trace, Site B; lower trace, Site C. The bottom trace shows repeated binding events with zero Cy3-Cy5 FRET and low (0.24) Cy3-Cy7 FRET, consistent with binding to a distal part of the Cy7-labeled motif similar to Site B. The difference in Cy3-Cy5 FRET could be due to heterogeneous phasing of the *hsp70* nucleosome as suggested by the Cy5-Cy7 FRET histogram in c. Black asterisks mark transient Cy3-only periods, potentially caused by ultra-short-range 1D diffusion on the nucleosome, resulting in a temporary loss of FRET. Green asterisks indicate binding events to non-cognate sites on the nucleosome. **c**, Pie charts showing, for all binding events at site A (left pie chart, N = 65), B (middle, N = 45) or C (right, N = 25), the fraction of events that were followed by a second binding to site A, B or C. For example, if a nucleosome is visited by GAF-DBD three times – first at Site A, second also at Site A, and third at Site B – then an A-A rebinding event and an A-B rebinding event have occurred on this nucleosome. **d**, Cy5-Cy7 FRET histogram for the *hsp70* nucleosome (N = 377).

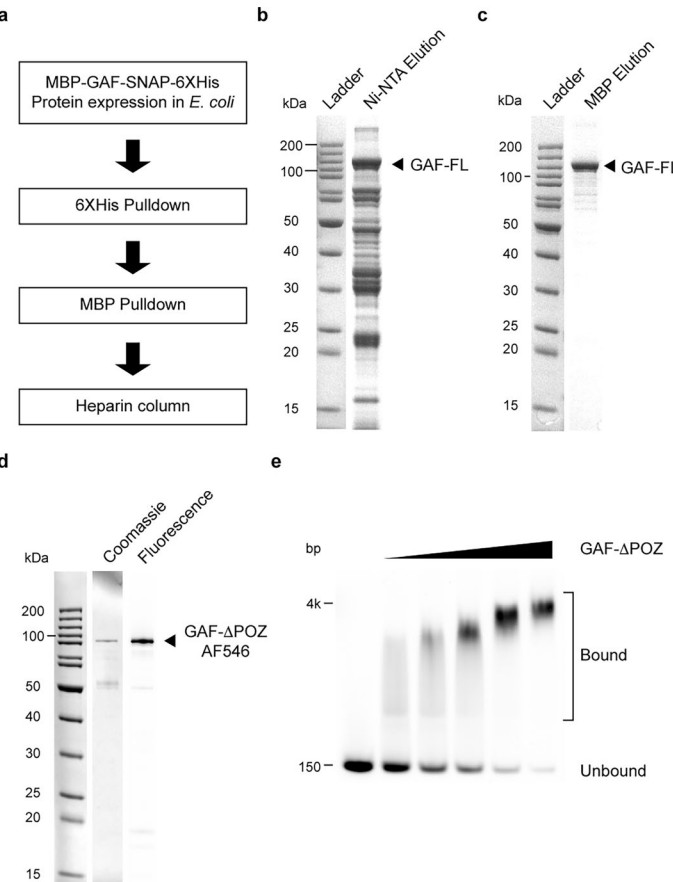

**Extended Data Fig. 6 | Purification of SNAP-tagged GAF-FL and GAF-ΔPOZ.**
**a**, GAF-FL purification workflow. **b**, SDS-PAGE gel of GAF-FL elution after 6XHis pulldown, stained with Coomassie Blue. **c**, SDS-PAGE gel of GAF-FL elution after MBP pulldown, stained with Coomassie Blue. **d**, SDS-PAGE gel of AF546-GAF-ΔPOZ, scanned for AlexaFluor 546 fluorescence, then stained with Coomassie Blue. **e**, EMSA for GAF-ΔPOZ binding to Cy5-labeled *hsp70* NCP DNA. GAF-ΔPOZ concentrations were 0, 4, 8, 12, 16, and 20 nM from left to right. DNA concentration was 3 nM. SDS-PAGE and EMSA gels shown in b-e were independently repeated once with similar results.

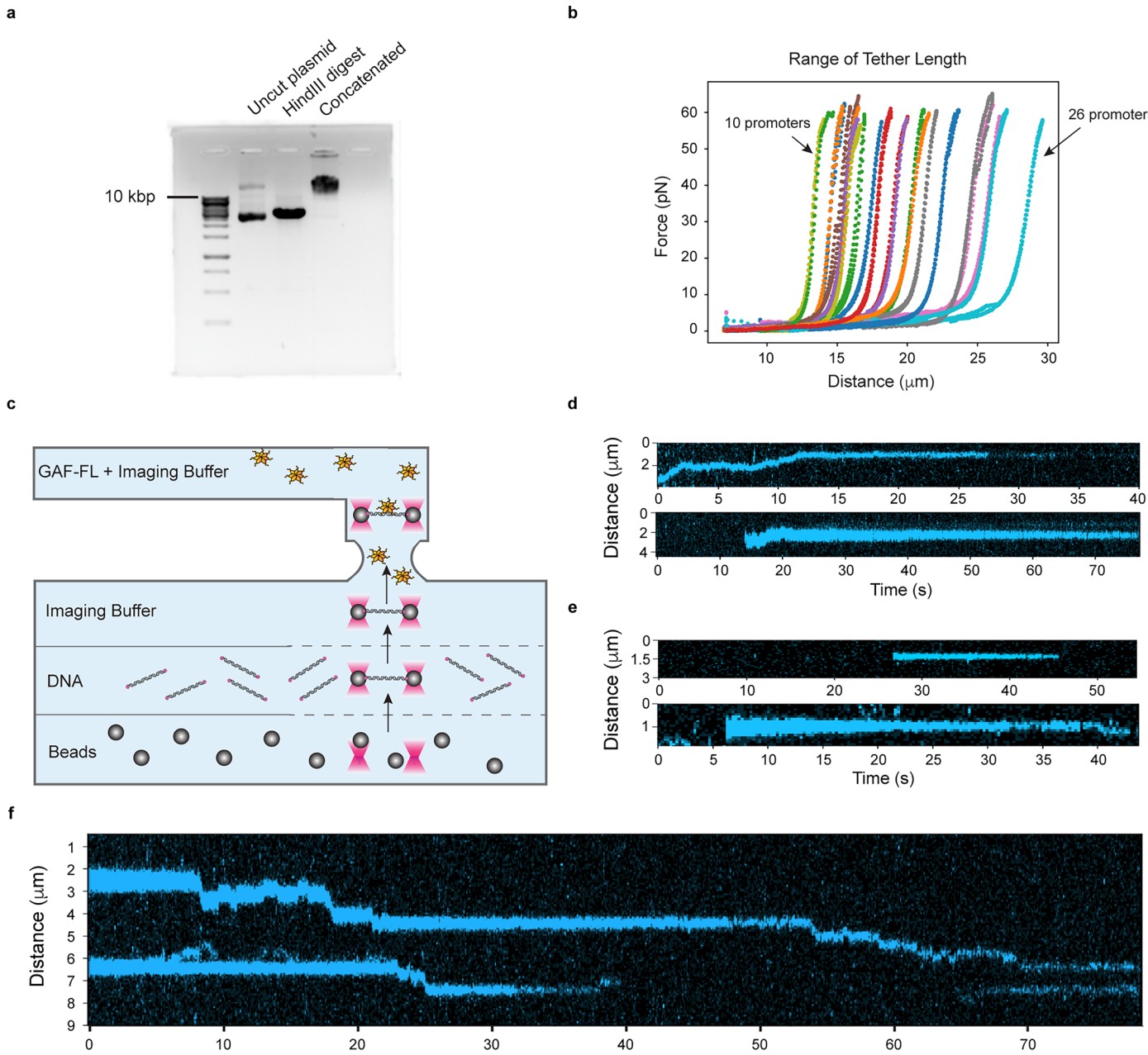

**Extended Data Fig. 7 | Optical tweezers experiment; design and confirmation of DNA assembly. a**, Agarose gel electrophoresis of concatenated plasmid DNA. This experiment was repeated once with similar results. **b**, Force versus distance plot reveals the length of double-stranded DNA tether. **c**, Diagram of flow cell constituents during imaging. Streptavidin coated beads, DNA, and Imaging Buffer were injected to the flow cell under laminar flow. Beads were first optically captured and moved to the DNA channel; once DNA was properly tethered to the trapped beads, the whole assemblage was moved to the protein channel containing AF-488 GAF-FL in Imaging Buffer. Imaging was performed in this channel to maximally visualize binding events. **d**, Representative kymographs where GAF-FL undergoes 1D search on vector+*hsp70* DNA. **e**, Representative kymographs where GAF-FL binds to its target on vector+*hsp70* DNA abruptly from 3D without 1D search. **f**, Representative kymograph where GAF-FL undergoes 1D diffusion from one target to another on vector+*hsp70* DNA.

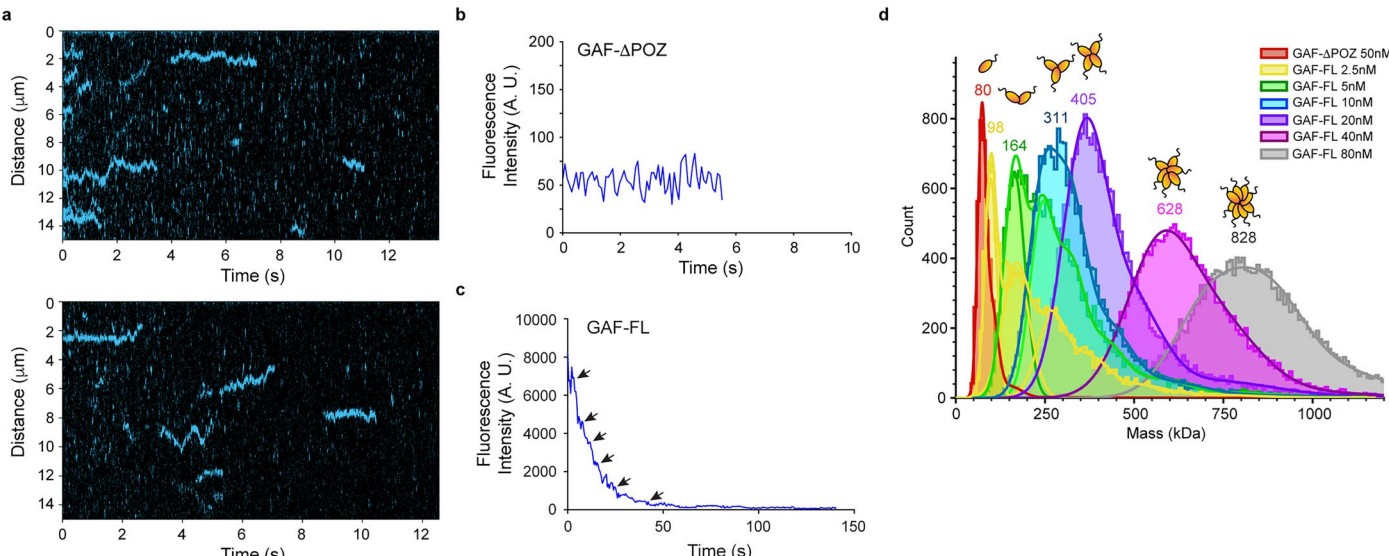

**Extended Data Fig. 8 | Behavior of full-length GAF depends on multimerization by POZ domain. a**, Representative kymographs show AF546-GAF-ΔPOZ binding to DNA over time on + *hsp70* DNA. **b**, Representative fluorescence intensity trace for GAF-ΔPOZ showing an abrupt loss of fluorescence signal at 5.7 s due to either dissociation or photobleaching. **c**, Representative trace showing GAF-FL photobleaching. Arrows indicate stepwise photobleaching. **d**, Mass photometry spectrum for GAF-ΔPOZ and GAF-FL constructs used in optical tweezer measurements. Spectra show the distribution of protein mass in a sample solution containing the indicated concentration of GAF. Notably, at similar protein concentrations as those used for the optical tweezer experiments (80 nM GAF-FL and 25 nM GAF-ΔPOZ), GAF-FL exists predominantly as an octameric multimer whereas GAF-ΔPOZ exists as a monomer.

# Reporting Summary

## Statistics

For all statistical analyses, confirm that the following items are present in the figure legend, table legend, main text, or Methods section.

| n/a | Confirmed | |
|---|---|---|
| ☐ | ☒ | The exact sample size (*n*) for each experimental group/condition, given as a discrete number and unit of measurement |
| ☐ | ☒ | A statement on whether measurements were taken from distinct samples or whether the same sample was measured repeatedly |
| ☐ | ☒ | The statistical test(s) used AND whether they are one- or two-sided<br>*Only common tests should be described solely by name; describe more complex techniques in the Methods section.* |
| ☒ | ☐ | A description of all covariates tested |
| ☒ | ☐ | A description of any assumptions or corrections, such as tests of normality and adjustment for multiple comparisons |
| ☐ | ☒ | A full description of the statistical parameters including central tendency (e.g. means) or other basic estimates (e.g. regression coefficient) AND variation (e.g. standard deviation) or associated estimates of uncertainty (e.g. confidence intervals) |
| ☐ | ☒ | For null hypothesis testing, the test statistic (e.g. *F*, *t*, *r*) with confidence intervals, effect sizes, degrees of freedom and *P* value noted<br>*Give P values as exact values whenever suitable.* |
| ☒ | ☐ | For Bayesian analysis, information on the choice of priors and Markov chain Monte Carlo settings |
| ☒ | ☐ | For hierarchical and complex designs, identification of the appropriate level for tests and full reporting of outcomes |
| ☒ | ☐ | Estimates of effect sizes (e.g. Cohen's *d*, Pearson's *r*), indicating how they were calculated |

*Our web collection on statistics for biologists contains articles on many of the points above.*

## Software and code

Policy information about availability of computer code

| Data collection | smFRET data were collected using smCamera2 (https://github.com/Ha-SingleMoleculeLab/smCamera2). Optical tweezers and confocal microscopy data were collected using BlueLake software (LUMICKS). |
|---|---|
| Data analysis | smFRET data were analyzed using custom-written IDL and MATLAB scripts (https://github.com/ashleefeng/gaf_target_search; https://github.com/Yibenfu/HMM_3colorFRET); Optical tweezers and confocal microscopy data were analyzed using Pylake v.1.0.0 (https://lumicks-pylake.readthedocs.io/en/v1.0.0/theory/diffusion/diffusion.html). Graphs were plotted using OriginLab (https://www.originlab.com). |

For manuscripts utilizing custom algorithms or software that are central to the research but not yet described in published literature, software must be made available to editors and reviewers. We strongly encourage code deposition in a community repository (e.g. GitHub). See the Nature Portfolio guidelines for submitting code & software for further information.

## Data

Policy information about availability of data

All manuscripts must include a data availability statement. This statement should provide the following information, where applicable:
- Accession codes, unique identifiers, or web links for publicly available datasets
- A description of any restrictions on data availability
- For clinical datasets or third party data, please ensure that the statement adheres to our policy

All data are available within this manuscript and its supplementary information files. Raw data in "PMA" format for smFRET and "H5" format for 2D confocal scans and kymographs are available from the corresponding authors upon request due to the large file sizes.

## Research involving human participants, their data, or biological material

Policy information about studies with human participants or human data. See also policy information about sex, gender (identity/presentation), and sexual orientation and race, ethnicity and racism.

| | |
|---|---|
| Reporting on sex and gender | N/A |
| Reporting on race, ethnicity, or other socially relevant groupings | N/A |
| Population characteristics | N/A |
| Recruitment | N/A |
| Ethics oversight | N/A |

Note that full information on the approval of the study protocol must also be provided in the manuscript.

# Field-specific reporting

Please select the one below that is the best fit for your research. If you are not sure, read the appropriate sections before making your selection.

☒ Life sciences  ☐ Behavioural & social sciences  ☐ Ecological, evolutionary & environmental sciences

For a reference copy of the document with all sections, see nature.com/documents/nr-reporting-summary-flat.pdf

# Life sciences study design

All studies must disclose on these points even when the disclosure is negative.

| | |
|---|---|
| Sample size | For single-molecule experiments, 100-300 trajectories were obtained for each condition, typically from at least 3 individual experiments. No statistical analysis was used to predetermine sample sizes, which were chosen based on our past experience with these experiments (Panja et al. PMID 28825710,  Hua et al. PMID 32913225). |
| Data exclusions | For single-molecule FRET experiments, traces were excluded if two or more peaks were evident. The exclusion criteria were pre-established in the field to address common technical issues such as optically overlapping molecules. |
| Replication | All experiments were performed at least 3 times in independent experiments. Replication was successful for all experiments. |
| Randomization | Not relevant to this study. |
| Blinding | Blinding was not relevant during data collection because our study did not involve human participants. Sample groups were not blinded during data analysis. |

# Reporting for specific materials, systems and methods

We require information from authors about some types of materials, experimental systems and methods used in many studies. Here, indicate whether each material, system or method listed is relevant to your study. If you are not sure if a list item applies to your research, read the appropriate section before selecting a response.

## Materials & experimental systems

| n/a | Involved in the study |
|---|---|
| ☒ | ☐ Antibodies |
| ☐ | ☒ Eukaryotic cell lines |
| ☒ | ☐ Palaeontology and archaeology |
| ☒ | ☐ Animals and other organisms |
| ☒ | ☐ Clinical data |
| ☒ | ☐ Dual use research of concern |
| ☒ | ☐ Plants |

## Methods

| n/a | Involved in the study |
|---|---|
| ☒ | ☐ ChIP-seq |
| ☒ | ☐ Flow cytometry |
| ☒ | ☐ MRI-based neuroimaging |

## Eukaryotic cell lines

Policy information about cell lines and Sex and Gender in Research

| | |
|---|---|
| Cell line source(s) | SF9 cell line was from the Eukaryotic Tissue Culture Facility at Johns Hopkins School of Medicine. |
| Authentication | SF9 cell line was not authenticated for this study. |
| Mycoplasma contamination | SF9 cells were not tested for mycoplasma contamination for this study. |
| Commonly misidentified lines (See ICLAC register) | N/A |

## Plants

| | |
|---|---|
| Seed stocks | N/A |
| Novel plant genotypes | N/A |
| Authentication | N/A |

