## [Peer Review File · Nature Structural & Molecular Biology]

GAGA zinc finger transcription factor searches chromatin by 1D-3D facilitated diffusion

Corresponding Author: Professor Carl Wu

Version 0:

Decision Letter:

10th Oct 2024

Dear Professor Wu,

Thank you again for submitting your manuscript "GAGA zinc finger transcription factor searches chromatin by 1D-3D facilitated diffusion". I sincerely apologise for the delay in responding while we obtained all necessary input and discussed it amongst the editorial team. We now have comments (below) from the 4 reviewers who evaluated your paper. In light of these reports, we remain interested in your study and would like to see your response to the comments of the referees, in the form of a revised manuscript.

You will see that while the reviewers appreciate the potential of the study, they nevertheless raise concerns about missing information and controls, and highlight the need for clearer reporting and improved clarity in different sections. More specifically, Reviewer #1 requests further discussion on non-specific binding events, and suggests further exploration of the question of direct binding of GAF from solution and the effect of a nucleosomal barrier on 1D diffusion, while Reviewer #2 asks for reporting of GAF concentrations, providing ensemble measurements to go along with the single-molecule results (echoed by reviewer #3), and provides suggestions to better streamline the data presentation. Reviewer #3 additionally requests further clarifications regarding the observed dwell times and Reviewer #4 also brings up ambiguous points and requests improvements to ensure the rigour and quality of the study. Please be sure to address/respond to all concerns of the referees in full in a point-by-point response and highlight all changes in the revised manuscript text file. If you have comments that are intended for editors only, please include those in a separate cover letter.

We expect to see your revised manuscript within 3 months. If you cannot send it within this time, please contact us to discuss an extension; we would still consider your revision, provided that no similar work has been accepted for publication at NSMB or published elsewhere.

Reporting Summary:

Data availability: this journal strongly supports public availability of data. All data used in accepted papers should be available via a public data repository, or alternatively, as Supplementary Information. If data can only be shared on request, please explain why in your Data Availability Statement, and also in the correspondence with your editor. Please note that for some data types, deposition in a public repository is mandatory - more information on our data deposition policies and available repositories can be found below:

<https://www.nature.com/nature-research/editorial-policies/reporting-standards#availability-of-data>

Nature Structural & Molecular Biology is committed to improving transparency in authorship. As part of our efforts in this direction, we are now requesting that all authors identified as 'corresponding author' on published papers create and link their Open Researcher and Contributor Identifier (ORCID) with their account on the Manuscript Tracking System (MTS), prior to acceptance. This applies to primary research papers only. ORCID helps the scientific community achieve unambiguous attribution of all scholarly contributions. You can create and link your ORCID from the home page of the MTS by clicking on 'Modify my Springer Nature account'. For more information please visit please visit www.springernature.com/orcid.

Link Redacted

Sincerely,

Dimitris Typas
Senior Editor
Nature Structural & Molecular Biology
ORCID: 0000-0002-8737-1319

Reviewers' Comments:

Reviewer #1 (Remarks to the Author):

This manuscript presents an extensive analysis of the GAF target search mechanism using multicolor single-molecule FRET, optical trapping, and confocal microscopy. The authors demonstrate that GAF combines 1D and 3D search mechanisms to locate binding sites in nucleosome-depleted regions and can frequently switch between neighboring binding sites on the same DNA molecule without fully dissociating. However, nucleosomes act as potent barriers to 1D diffusion, and recognition of nucleosomal binding sites largely relies on direct 3D binding. To my knowledge, this manuscript represents the first analysis of this kind for a eukaryotic transcription factor and highlights intriguing differences from prokaryotic transcription factors, which do not need to operate in the context of largely nucleosome-covered DNA. These results represent a significant step forward in the mechanistic understanding of eukaryotic transcription factor target search.

The data are of outstanding quality and are presented clearly. Therefore, I enthusiastically recommend publication of this manuscript, after the following points have been addressed. While I have suggested a small number of additional experiments that could further strengthen the manuscript, I believe that all of my points could be addressed textually.

Major points:

The authors have observed that the frequency of binding events (and thus k_{on}) is 16-fold higher for cognate versus non-cognate DNA sequences. This is an unexpected result that suggests most specific binding events occur directly, without the non-specific binding stage. It would seem to put a low ceiling on the potential effects of facilitated diffusion and also contradicts other observations in the manuscript, namely that around 20% of specific binding events in 2-color FRET experiments and 40-50% of binding events in 3-color FRET experiments are preceded by non-specific binding. These analyses rely on the assumption that the temporal resolution of these experiments is sufficient for reliably detecting the majority of non-specific binding events. However, based on data in Fig 1f, it seems like a large fraction of non-specific binding events last less than one frame. Additionally, the 1D diffusion coefficient for monomeric GAF suggests that it should encounter the binding site much faster during 1D diffusion. One potential explanation could be that the TF on average needs to encounter the binding site many times before specific binding, but that would argue against efficient direct binding from solution. The authors should discuss these apparent discrepancies in more detail.

The high apparent probability of direct binding from solution observed in at least some of the experiments is one of the most striking features of GAF. However, given the limited time resolution of single-molecule FRET experiments, the authors should consider further exploring this question. Could they directly probe 3D binding under conditions when 1D diffusion is minimized? For example, a comparison of binding kinetics in a bulk or a single-molecule experiment using a DNA molecule that does not have extra non-specific DNA (if it is too short to remain stable in solution, a hairpin can be used), or where binding from extra DNA is prevented by roadblocks or other means versus the case where 1D diffusion is allowed, could substantially strengthen the manuscript. Alternatively, the authors could textually address these limitations.

The assay developed by the authors enables an in-depth analysis of the GAF target search. Given the observation that a nucleosome serves as a barrier for 1D sliding, this assay provides a unique opportunity to probe intersegment transfer by positioning two binding sites on opposite sides of a nucleosome. The authors could further strengthen the manuscript by directly characterizing this elusive step in the target search process, although such an experiment is clearly not required to support the authors' conclusions.

Minor points:

Unlabeled protein molecules can often represent a challenge for single-molecule analysis of site-specific interactions by occupying binding sites without being observable. In this case, the very high labeling efficiency of GAF-DBD (92%) makes this issue irrelevant. However, it would be instructive to state somewhere in the text that unlabeled GAF molecules did not present a problem due to high labeling efficiency.

The statements about the ability to distinguish photobleaching versus dissociation of monomeric GAF (line 311 and EDF9 b-c) are not sufficiently well explained. It appears that the criterion for this distinction was the level of residual fluorescence after photobleaching, but it is not clear to me what the nature of this residual fluorescence is, and whether photobleaching would always result in detectable residual fluorescence.

Line 316: when comparing diffusion coefficients of monomeric and multimeric GAFs, it would be useful to mention that a substantial difference in diffusion coefficients is expected purely based on the difference in size.

Line 298: given the limited resolution of tracking, it would be hard to distinguish short-range 1D diffusion from direct 3D binding. This point is mentioned in the discussion, but it would make sense to mention it here too and state the resolution of tracking.

Some methods sections are missing intended references, having (Ref) marks instead, e.g., lines 891, 950, 952, and 962.

EDF 9: typos in panels b and e ('photobleaching').

EDF 4 and 5: it seems excessive to provide percentages up to the first decimal place, especially for cases with $N < 100$.

Line 43: should probably read "...transcription factors such as LacI SEARCH efficiently for THEIR TARGETS..."

Reviewer #2 (Remarks to the Author):

The manuscript by Feng et al. reports single-molecule fluorescence studies of the *Drosophila* transcription factor, GAGA-Associated Factor (GAF) interacting with DNA and nucleosomes containing GAF target sequences. Native full-length GAF (GAF-FL) functions as a multimer with an average of 6 monomers. While the authors include studies of GAF-FL, they largely focus on a GAF mutant where much of the protein is deleted so it contains the minimal length for site-specific DNA binding. They refer to this protein, which functions as a monomer, as the GAF DNA binding domain (GAF-DBD). In addition, they study GAF with the POZ domain deleted, which abolishes multimerization. This results in GAF functioning as a monomer,

which they refer to this construct as GAF- POZ.

Most of the measurements in the manuscript study fluorophore-labeled GAF-DBD with single molecule Total Internal Reflection Microscopy (smTIRF). They label GAF-DBD with Cy3 so that it undergoes efficient FRET with a fluorophore attached just outside the DNA target site. They studied both one site using Cy5, and two sites using Cy5 and Cy7. This elegantly allows them to observe both GAF-DBD binding to the DNA and when GAF-DBD is at one of the binding sites. They use different DNA constructs. One contains part of the native sequence from hsp70 promoter and another contains the Widom 601 high-affinity nucleosome positioning sequence. They carried out smTIRF measurements with a range of different constructs. Their main conclusion is that (i) GAF-DBD undergoes a combination of 1D and 3D diffusion to bind its target sequence and if there are two sites it slides back and forth along the DNA between the two sites, and (ii) GAF-DBD 1D diffusion is blocked by nucleosomes so that it relies on 3D diffusion to access sites within the nucleosome.

The authors also provide gel shift data to show that GAF-FL can bind to its target sequence within nucleosomes similar to free DNA. This supports the conclusion that GAF-FL functions as a pioneer factor. Finally, the authors provide C-Trap measurements of GAL-FL sliding along DNA tethers with multiple target sequences spaced semi-randomly with kb spacing. These results show GAF-FL can slide over kb distances and target multiple sites through 1D diffusion.

This is an extensive single-molecule study that address an important question, namely how do site-specific DNA binding transcription factors find their target sequences within chromatin? Their approach of using a combination of colocalization and FRET to determine both DNA binding and binding to a target sequence within the DNA molecule is elegant and insightful. These are the central findings of the work. The gel shift assay that shows GAF-FL can target its site efficiently within nucleosomes and the C-Trap measurements on long DN molecules provide additional interest information. However, these last two measurements are not that well connected to the previous studies of GAL-DBD.

This is an interesting study with a significant amount of high-quality single-molecule (smTIRF) measurements of GAF. In addition, the study includes some gel shift and C-Trap measurements. But they are a bit disconnected from the smTIRF measurements, which causes the study to be a bit disjointed. However, overall, these studies provide important insight into how GAF targets sites within DNA and how nucleosomes influence monomeric GAF.

Below I provide feedback to help improve this study.

1. The authors report binding rates for the smTIRF and C-Trap measurements. However, they do not report (at least I could not find) the concentration of the GAF protein. The binding rates depend on the concentration, which is why one typically reports a binding rate constant in units of $\text{time}^{-1}\text{concentration}^{-1}$. The authors should include the GAF concentrations used in each experiment and include the binding rate constants in addition to the binding rates. Also, ideally, binding measurements are done at different concentrations to show the expected linear dependence of binding rate on concentration.
2. Related to point 1, the authors do not provide ensemble measurements of the apparent KD. This could be measured by electromobility gel shift assays (EMSAs) or by fluorescence. I assume they did these measurements to confirm the activity of their GAF samples. I strongly suggest they include EMSA measurements of GAF-DBD and GAF- POZ.
3. Related to point 1 and point 2, it would be good to compare the rate measurements to the apparent KD of GAF-DBD. The authors just need to divide the binding rate of GAF-DBD by the concentration used to determine the binding rate constant. They can then divide the dissociation rate by the binding rate constant and compare this to the apparent KD. These do not necessarily need to be the same, but they should be within the same order of magnitude.
4. The authors provide the cumulative distribution functions (CDF) for some of their dwell time measurements (Figures 4 and 5) but not for others. The Dwell times reported in Figures 1, 2, and 3 most likely come from CDFs. The authors should provide the CDFs for each effective dwell time that is determined from a distribution of dwell times with the fits used to determine the overall dwell time.
5. There is a large number of different dwell times and rates reported. It is hard to keep track of all of them and to see how they compare. To help, the authors should provide a supplementary table with all the determined rates/dwell times with their uncertainty.
6. It is interesting the binding rate to DNA with the GAF-DBD target site (cognate DNA) is higher than to DNA without the target site (non-cognate DNA). This suggests that there are binding events to the non-cognate DNA that are too short to observe since it is unlikely that when GAF-DBD is not bound to DNA that it "knows" if the DNA has a binding site or not. The authors should discuss this more in the Results and/or Discussion sections.
7. It is challenging to keep track of the different sites and the distance between sites. It would be very helpful to provide names of the binding sites, to indicate them in the figure diagrams, and to provide the base pair spacing within the diagrams. This should be done by adding a diagram in Figure 1 and adding this information to Figures 2A, 3A, and 4A.
8. Similar to point 7, the authors should provide diagrams of the three protein constructs they use: GAF-DBD, GAF-FL, and GAF- POZ.
9. For the 3-color FRET measurements in Figure 4, there could be two separate GAF-DBD binding to each site. The traces that show the Cy5 and Cy5 fluorescence increase simultaneously are very likely one GAF-DBD. However, there are likely

traces with 2 GAF-DBD bound. The authors should be clearer about whether there are events with 2 GAF-DBD bound? If so, what is the fraction of these events? Alternatively, if not, then why are these not observed?

10. It is interesting and perhaps surprising that the overall dwell time on the DNA fragment with 2 sites is similar to the dwell time on the 90 bp DNA with one binding site. Perhaps this is because the second site has a lower affinity sequence. The authors use a DNA construct later with 2 high-affinity sites. But they only study this DNA construct wrapped into nucleosomes. Did the authors happen to do DNA-only studies with the DNA construct with two high-affinity sites? It would be interesting to determine if the resident time is significantly increased.

11. The authors report where the binding site within the nucleosome is located relative to the superhelical location (SHL). Is this based on the center of the binding site? It would be helpful if the authors also reported how far the binding site extends into the nucleosomes.

12. For the measurements with nucleosomes, how do the authors know that their molecules contain a full histone octamer after surface tethering? Is it possible that some of the molecules have lost one or both histone heterodimers? The authors should discuss this possibility.

13. The authors use abbreviations that are not clearly defined in the manuscript (at least I could not find definitions). This includes CDF, which is a cumulative distribution function. Also, the authors refer to A-A, B-B, and C-C events, but these events do not appear to be defined. Please define all abbreviations in the manuscript.

14. The authors provide SDS PAGE gels of the different GAF protein samples. However, there is no characterization of the reconstituted nucleosomes. Typically, this is done with Electromobility Shift Assays (EMSAs). The authors should provide EMSAs of the different nucleosome constructs used in the different single-molecule assays.

15. The methods references have issues that need to be addressed. In some places, the text has (Ref). I assume this is a placeholder and the authors forgot to insert the correct reference. In other places, the authors list the last name of the first author and the year. Each reference should have a full reference that is listed at the end of the methods.

16. More details in the Methods section need to be provided. They provide excellent detail on the preparations of GAF-DBD and GAF. However, details on the histone octamer and nucleosome preparations are mostly lacking. The authors should not completely rely on referencing previous papers. Please provide in the Methods section more details on how the histone octamer was refolding and purified, and how nucleosomes were reconstituted and purified.

Reviewer #3 (Remarks to the Author):

Feng et al. present a single molecule study of GAGA-associated factor (GAF) binding and diffusion on DNA and nucleosomal templates. Both the DNA binding domain and the full-length GAF can engage in 1D diffusion on naked DNA, which facilitates target search. Nucleosome blocks sliding beyond the entry / exit sites, and solvent-exposed motifs on the nucleosomes are mostly accessed through direct binding.

While the findings may not be groundbreaking, this paper offers a valuable contribution to the field. GAF plays a crucial role in gene regulation and development, particularly as a pioneer factor, making its interaction with nucleosomes highly relevant from a physiological perspective. The use of both the 601 nucleosome positioning sequence and endogenous GAF binding sequence demonstrates a rigorous approach. The diffusion data obtained through 3-color FRET and C-trap methods are particularly intriguing.

There are several points that require further clarification:

1. Is the data in Figure 1 measured at one concentration of GAF-DBD? If yes, how is this concentration determined? Does the single molecule data roughly agree with bulk measurement of K_d ?
2. Figure 1g & h. The error bars for the "technical replicates" appear large. How were these calculated? The number of traces and binding events should be reported for accuracy.
3. The average dwell time of GAF in Figure 1 seems significantly longer than that on Sites 1 and 2 in Figure 2, despite Site 2 having the same motif. What explains this difference? If it's due to entrapment at Site 1, what is the average dwell time on Site 2 after mutating Site 1 (see Extended Data Figure 4c-e)?
4. Figure 3 only presents the binding site oriented towards the histone. It would be beneficial to show some traces of SHL2.5, 4.5, and 6.5, ideally also the corresponding rastergrams. The extensive scanning at SHL4.5 (significantly more than SHL6.5 according to Extended Data Figure 5) is surprising and warrants an explanation.
5. I am not fully convinced of the data interpretation in Figure 4C. The diagram in the middle (dyad site distal to Cy7) should lead to decreased FRET efficiency of Cy5 and Cy7, but the traces above do not show such trend. Some explanation should be provided. Maybe the Cy7 motif can be shortened to eliminate the possibility of multi-mode binding.
6. Related to question 3 above, the dwell time on the same motif but in different templates seem to vary significantly. I thus wonder if the comparison in Figure 4g is fair. How about the dwell time of GAF on 601 nucleosome at SHL2.5, 4.5, and 6.5? Do they also show longer dwell time?
7. Current Figure 4 is somewhat confusing because the data in panels a-g are based on GAF DBD, whereas panel h uses the full-length GAF. Including an EMSA comparison between nucleosome and naked DNA using GAF DBD would enhance clarity.

Minor:

In Figure 2c, the diagram appears to indicate "50" rather than "55".
It would be more informative to show 1-CDF using semi-log plot.
Figure 4h is not cited correctly in the manuscript; it is referred to as Extended Data Figure 6 instead.

Reviewer #4 (Remarks to the Author):

This manuscript presents significant scientific findings regarding how eukaryotic sequence-specific transcription factors (TFs) search for gene targets on chromatin, influenced by nucleosomes. The research employs state-of-the-art approaches, making the discoveries relevant for publication in NSMB. I generally support publication but have suggestions for improving the manuscript's quality and clarity.

1. Clarification of 1D vs. 3D Search: The physiological significance of 1D or 3D search modes remains unclear. The authors need to explain why this behavior is important. It would be helpful to focus on the observations in the model shown in Extended Data Figure 10. Speculation should be minimized. The authors should emphasize the data and clarify the functional differences between 1D and 3D searching mechanisms.

2. Multimeric Status of the TF: The multimeric status of the transcription factor requires more clarity. Although the authors suggest an average hexameric state, the data in Extended Data Figure 9d do not convincingly support six photobleaching steps. Moreover, the intensity differences between Extended Data Figure 9b and 9d are quite large although the intensity unit is arbitrary. I recommend comparing the intensities of a single spot with single photobleaching step intensities for a more quantitative analysis. If there are no structural studies supporting the hexameric state, the possibility that the multimeric status is a condensate should be considered, as many TFs display condensation properties. A more quantitative description is needed to clarify the function of this multimeric state.

3. 1D Diffusion Mechanism: It would be helpful if the authors structurally explain how the GAF DBD or its multimeric state can perform 1D diffusion. Does GAF DBD possess a DNA anchoring site that enables it to slide along DNA? Alternatively, could there be a pore created by the assembly of GAF protomers?

4. Hopping vs 1D diffusion in FRET Data: The authors should clarify whether the FRET results exclude the possibility of hopping events, which are commonly observed during DNA-protein interactions. If hopping is not occurring, the authors should rule out this possibility in the manuscript. Additionally, for 1D diffusion, a gradual change in FRET values would be expected. Please provide this data.

5. Cy5 and Cy7 Data Interpretation: In the Cy5 and Cy7 data, where the motif is positioned at the nucleosome dyad and edge, there are instances where only one signal lights up and cases where both signals light up simultaneously. This raises the question of whether there are three distinct binding modes or if the wide variation in Cy5 to Cy7 signal ratios suggests a more dynamic binding mode. The FRET values should be provided to clarify whether the GAF-DBD bound state is a single state or multiple intermediates.

6. Transition Between Cy5 and Cy7 Signals: The manuscript shows a Cy3-signal recovery during transitions between Cy5 and Cy7 signals, which suggests sliding between two binding sites on DNA. However, there are cases where Cy5 and Cy7 transitions occur directly, which may not indicate sliding. Do the authors have statistical data on the ratio of direct transitions versus Cy3-mediated transitions? If there are missed Cy3 detections during Cy5 to Cy7 transitions, it would be helpful to address this limitation.

7. Conversion of Length to Base Pairs: The conversion between length and base pairs may vary depending on the force applied in the optical tweezer experiments. Was this calculation based on an assumed value of 0.34 nm per base pair? Additionally, the authors should clarify if the binding between DNA and protein changes depending on the tension applied to the DNA.

8. Kymographs in Figure 5c,d: The kymographs in Figure 5c,d show changes in protein intensities. The authors need to clarify whether these changes are due to photobleaching or the dissociation of multimeric states.

9. Cy3 Donor-only Fluorescence: In line 112 on page 3, the authors should clarify that the transient Cy3 donor-only fluorescence flanked by FRET events is not a result of photo-blinking.

10. Binding Time and 1D Sliding: In the early section, the manuscript mentions that GAF-DBD remains bound to the motif about 2% longer than the theoretical binding time calculated based on 3D diffusion. If the theoretical 3D binding probability is 66%, does this mean that 60 bp of the 90 bp DNA used in the experiment consists of the GAGAG motif, or is this a thermodynamic probability based on binding energy in 3D? Clarification is needed.

11. Rebinding Tendency: In Extended Data Figure 6, it is noted that dissociation and re-binding of the same type (A-A, B-B, C-C) occur more than 80% of the time. Could this observation be influenced by nucleosome assembly differences? The authors should clarify whether similar trends are observed with longer interval times before re-binding.

12. 1-CDF Data Comparison: The graph calculated with 1-CDF shows that motif-specific dwell time differs tenfold between

DNA and nucleosomes. Is this comparison based on maximum values, or does it refer to a comparison of average data? Clarification would improve interpretation.

13. GAF-FL Binding Comparison: In line 276, the manuscript compares GAF-FL binding to NCP versus bare DNA via titration, noting that NCP binds about half as much as bare DNA. Could the calling in Extended Data Figure 6 be double-checked to ensure accuracy?

14. Typo: In line 307, please delete “, d”.

I believe that addressing these points will significantly strengthen the manuscript's quality and scientific rigor.

Version 1:

Decision Letter:

Our ref: NSMB-A49286A

26th May 2025

Dear Professor Wu,

Thank you for submitting your revised manuscript "GAGA zinc finger transcription factor searches chromatin by 1D-3D facilitated diffusion" (NSMB-A49286A). It has now been seen by the original referees and their comments are below. The reviewers find that the paper has improved in revision, and therefore we'll be happy in principle to publish it in Nature Structural & Molecular Biology, pending minor revisions to satisfy the referees' final requests and to comply with our editorial and formatting guidelines.

We are now performing detailed checks on your paper and will send you a checklist detailing our editorial and formatting requirements in about 1-2 weeks. Please do not upload the final materials and make any revisions until you receive this additional information from us.

To facilitate our work at this stage, it is important that we have a copy of the main text as a word file. If you could please send along a word version of this file as soon as possible, we would greatly appreciate it; please make sure to copy the NSMB account (cc'ed above).

Sincerely,

Dimitris Typas
Senior Editor
Nature Structural & Molecular Biology
ORCID: 0000-0002-8737-1319

Reviewer #1 (Remarks to the Author):

The authors have addressed all points thoroughly, and I enthusiastically recommend publication of the manuscript.

Reviewer #2 (Remarks to the Author):

The authors have addressed all of the reviewer comments. I have no additional feedback to provide.

Reviewer #3 (Remarks to the Author):

The authors did a good job addressing all my concerns. I support the publication of the manuscript.

Reviewer #4 (Remarks to the Author):

The reviewer's comments have provided valuable and insightful feedback that substantially enhances the scientific clarity and overall quality of the manuscript. In particular, the review has helped improve the work in three major areas: data support, interpretational clarity, and structural insight.

First, the reviewer encouraged the addition of quantitative and orthogonal evidence to support the multimeric state of the transcription factor. In response, the authors performed mass photometry experiments under the same conditions used in the

single-molecule imaging setup, confirming that GAF-FL forms predominantly hexamers to octamers in a concentration-dependent manner. This new dataset was essential for validating the multimerization behavior and is now included in Extended Data Figure 9f.

Second, the reviewer raised important points regarding the functional differences between 1D and 3D search mechanisms, as well as the distinction between hopping and rotation-coupled sliding during 1D diffusion. These suggestions led to clearer theoretical framing of diffusion behaviors, a more grounded discussion of physiological relevance (now clarified in lines 318–337 and 397–403), and new experiments testing the salt-dependence of the nonspecific binding state. These buffer modulation experiments revealed that lower ionic strength increases dwell time, supporting the presence of hopping behavior during 1D diffusion.

Third, the reviewer asked for a deeper structural explanation of how the GAF DNA-binding domain (DBD) engages with DNA to support diffusion. In response, the authors incorporated prior structural data showing that GAF-DBD contains a zinc finger flanked by three basic regions that mediate both major and minor groove as well as backbone interactions—mechanistically supporting a model for nonspecific DNA scanning.

Additional suggestions, such as the interpretation of FRET transitions, nucleosome phasing, kymograph intensity decay, and theoretical calculations for binding probability and sliding range, were all addressed with either new quantitative analysis, clarifications in the main text, or contextual explanations referencing relevant literature. Wherever appropriate, the authors also acknowledged the limitations of their data (e.g., time resolution constraints for detecting gradual FRET changes during sliding), demonstrating scientific transparency and rigor.

Overall, the reviewer's detailed comments have led to substantial improvements in the logic, presentation, and depth of the manuscript. The authors have addressed the concerns systematically with experimental evidence and mechanistic discussion, and the revised version is now scientifically robust and clearly articulated. These revisions not only strengthen the manuscript but also expand its broader relevance to the field of chromatin-associated TF dynamics.

Version 2:

Decision Letter:

1st Jul 2025

Dear Professor Wu,

We are now happy to accept your revised paper "GAGA zinc finger transcription factor searches chromatin by 1D-3D facilitated diffusion" for publication as an Article in Nature Structural & Molecular Biology.

Your paper will be published online soon after we receive proof corrections and will appear in print in the next available issue. You can find out your date of online publication by contacting the production team shortly after sending your proof corrections.

Sincerely,

Dimitris Typas
Senior Editor
Nature Structural & Molecular Biology
ORCID: 0000-0002-8737-1319

Responding to:

Reviewer #1 (Remarks to the Author):

This manuscript presents an extensive analysis of the GAF target search mechanism using multicolor single-molecule FRET, optical trapping, and confocal microscopy. The authors demonstrate that GAF combines 1D and 3D search mechanisms to locate binding sites in nucleosome-depleted regions and can frequently switch between neighboring binding sites on the same DNA molecule without fully dissociating. However, nucleosomes act as potent barriers to 1D diffusion, and recognition of nucleosomal binding sites largely relies on direct 3D binding. To my knowledge, this manuscript represents the first analysis of this kind for a eukaryotic transcription factor and highlights intriguing differences from prokaryotic transcription factors, which do not need to operate in the context of largely nucleosome-covered DNA. These results represent a significant step forward in the mechanistic understanding of eukaryotic transcription factor target search. The data are of outstanding quality and are presented clearly. Therefore, I enthusiastically recommend publication of this manuscript, after the following points have been addressed. While I have suggested a small number of additional experiments that could further strengthen the manuscript, I believe that all of my points could be addressed textually.

Major points:

1. The authors have observed that the frequency of binding events (and thus k_{on}) is 16-fold higher for cognate versus non-cognate DNA sequences. This is an unexpected result that suggests most specific binding events occur directly, without the non-specific binding stage. It would seem to put a low ceiling on the potential effects of facilitated diffusion and also contradicts other observations in the manuscript, namely that around 20% of specific binding events in 2-color FRET experiments and 40-50% of binding events in 3-color FRET experiments are preceded by non-specific binding. These analyses rely on the assumption that the temporal resolution of these experiments is sufficient for reliably detecting the majority of non-specific binding events. However, based on data in Fig1f, it seems like a large fraction of non-specific binding events last less than one frame.

We thank the reviewer for this comment. We agree that, as the reviewer pointed out, many non-specific binding events that we can detect are less than one frame (with only one data point and lower than average intensity value). This suggests that we may fail to detect many non-specific binding events that are even more transient. Therefore, we don't claim that the true k_{on} for GAF-DBD is different between our non-cognate and cognate DNA constructs. On the other hand, the measurable, apparent k_{on} is clearly distinguishable. This limitation may also apply to some specific binding events where the initial search phase is too short to be detected. Therefore, the 20% and 40-50% chance of detecting a search phase in our two- and three-color experiments, respectively, are likely underestimating the true propensity of GAF-DBD to slide on DNA before reaching the cognate motif. The true probability for our GAF-DBD undergoing a

search phase before specific binding should depend on the probability of the protein landing on non-specific DNA and may be close to the fraction of the non-cognate portion out of the entire DNA (~98%), which is much higher than what we observed. While we believe part of the discrepancy is because some protein molecules directly bind the cognate target from solution, we are likely missing a large number of nonspecific binding events at least under default buffer conditions (also see our response to the third point). We have modified the text to make this point clear (Lines 124-125).

2. Additionally, the 1D diffusion coefficient for monomeric GAF suggests that it should encounter the binding site much faster during 1D diffusion. One potential explanation could be that the TF on average needs to encounter the binding site many times before specific binding, but that would argue against efficient direct binding from solution. The authors should discuss these apparent discrepancies in more detail.

We thank the reviewer for noticing that the 1D diffusion coefficient for monomeric GAF Δ POZ as measured by C-trap appears higher than you would expect from the binding site switching frequency from our FRET data. There are three potential explanations for this: first, as the reviewer pointed out, monomeric GAF may bypass the cognate site several times before binding. Our data does suggest efficient direct binding but we cannot rule out cognate site bypassing because many apparent direct binding events may be preceded by a transient, undetectable search period, making direct binding potentially less efficient than it appears (also see our response to the previous and the next question).

Second, many transitions occurred directly from Cy3-Cy5 FRET to Cy3-Cy7 FRET or vice versa, without going through a Cy3-only period. In these cases, 1D diffusion between two cognate sites was too short to be captured under our time resolution. This may give rise to underestimation of the nonspecific state duration, or the transit time between the two motifs. The true 1D diffusion constant would thus be higher than what can be inferred from the Cy3-only state dwell time and be closer to the GAF Δ POZ diffusion constant.

The third possibility is that monomeric GAF may have two distinct diffusion modes, a slower mode near its cognate sites and a faster mode on nonspecific DNA. The 1D diffusion coefficient was measured by confocal microscopy where the spatial resolution (75 nm) (PMID 38497611) is diffraction limited and the DNA is 10-30 μ m; whereas our FRET assay detects movements on a DNA shorter than 70 nm. Thus our FRET setup might be much better at capturing short-range movements when the protein is near its cognate site whereas the C-trap is better at measuring long-range search behavior on nonspecific DNA (cognate DNA only makes up ~1% of the C-trap DNA).

Due to these limitations, while our 3-color FRET data is useful for visualizing short-range sliding and characterizing the kinetics for cognate site binding, calculating a 1D diffusion coefficient from this data can be misleading (hence we only report diffusion coefficients from C-trap measurements).

3. The high apparent probability of direct binding from solution observed in at least some of the experiments is one of the most striking features of GAF. However, given the limited time resolution of single-molecule FRET experiments, the authors should consider further exploring this question. Could they directly probe 3D binding under conditions when 1D diffusion is minimized? For example, a comparison of binding kinetics in a bulk or a single-molecule experiment using a DNA molecule that does not have extra non-specific DNA (if it is too short to remain stable in solution, a hairpin can be used), or where binding from extra DNA is prevented by roadblocks or other means versus the case where 1D diffusion is allowed, could substantially strengthen the manuscript. Alternatively, the authors could textually address these limitations.

We appreciate the reviewer's comment and agree that it is intriguing that a significant fraction of GAF-DBD and full-length GAF appear to directly bind cognate sites from solution for both linear DNA and nucleosomal DNA.

For linear DNA, the reviewer pointed out that the time resolution of our current methods limit our ability to resolve transient nonspecific binding and directly probe 3D binding. While this is true for our FRET data collected under default salt concentrations (50 mM NaCl, 6.25 mM MgCl₂) where the detected nonspecific binding (56 ms) approaches our 35 ms exposure time, direct binding still contributed 40% of total binding events when MgCl₂ concentration was lowered to 3 mM (Extended Fig. 4i) where 1D diffusion was significantly slowed down and nonspecific binding increased to 140 ms—making most nonspecific events detectable. Further, our C-trap data show that full-length GAF multimers appear to directly locate their targets in 44% of binding events, which cannot be fully explained by a preceding non-detectable 1D diffusion (lines 294-296 and Methods 1131-1140). Therefore, although our measurements likely overestimate the true fraction of direct binding, we believe that our data cannot be explained by 1D diffusion alone and suggest both GAF-DBD and full-length GAF can directly bind specific sites from solution.

The reviewer suggested using a very short DNA molecule (<10 bp) that does not have extra non-specific DNA and only allows 3D binding. We appreciate this suggestion and believe this would allow for more quantitative measurement of 3D binding, but this is not necessary given our data already provide sufficient evidence for 3D binding. This experiment is also challenging because it requires k-on measurements for unstable 5-bp DNA with ideally microsecond time resolution. We think perfecting the kinetic measurement of direct binding on cognate sites is important but not essential for our claim that there is a non-trivial contribution of 3D binding in our system.

For nucleosome binding, we can infer that the binding events observed for cognate motifs on nucleosomes are direct binding and not preceded by 1D diffusion. For GAF-DBD binding to the same DNA construct, alternating Cy5/Cy7 FRET was observed on the linear DNA but was abolished in the reconstituted nucleosome which fully covers the Cy7-labeled cognate site. This is also consistent with our data on 601 nucleosomes (Fig. 3 and Extended Data Fig. 5) where we no longer observe 1D diffusion when cognate motifs are located inside SHL6.5. Together our data suggest that GAF-DBD

cannot use 1D diffusion to locate the Cy7-labeled cognate motif and thus the specific events we observe for the Cy7-labeled inner nucleosomal motif (Fig 4d,e,f) come from direct binding from solution. For full-length GAF, our gel shift data using a nucleosome core particle containing almost no linker DNA—minimizing 1D diffusion—also suggest that GAF can directly bind nucleosomal DNA from solution. Together, our nucleosome binding experiments allow us to directly probe 3D binding and our data suggest targeting cognate motifs that lie inside a nucleosome (in the central 130 bp) require direct binding from solution.

4. The assay developed by the authors enables an in-depth analysis of the GAF target search. Given the observation that a nucleosome serves as a barrier for 1D sliding, this assay provides a unique opportunity to probe intersegment transfer by positioning two binding sites on opposite sides of a nucleosome. The authors could further strengthen the manuscript by directly characterizing this elusive step in the target search process, although such an experiment is clearly not required to support the authors' conclusions.

We thank the reviewer for this suggestion. Intersegmental transfer is an intriguing process to study but we think it would be best explored in a separate study. GAF binding sites usually do not span more than a nucleosome length of DNA (70 bp in the hsp70 promoter) so it is unclear whether GAF is the best model protein for studying this problem. We concur with the reviewer that this information is not required to support our conclusions.

Minor points:

5. Unlabeled protein molecules can often represent a challenge for single-molecule analysis of site-specific interactions by occupying binding sites without being observable. In this case, the very high labeling efficiency of GAF-DBD (92%) makes this issue irrelevant. However, it would be instructive to state somewhere in the text that unlabeled GAF molecules did not present a problem due to high labeling efficiency.

We thank the reviewer for this suggestion. We have now stated this in the text (lines 119-120).

6. The statements about the ability to distinguish photobleaching versus dissociation of monomeric GAF (line 311 and EDF9 b-c) are not sufficiently well explained. It appears that the criterion for this distinction was the level of residual fluorescence after photobleaching, but it is not clear to me what the nature of this residual fluorescence is, and whether photobleaching would always result in detectable residual fluorescence.

We thank the reviewer for this comment. We do not know what the nature of the residual fluorescence after photobleaching is, although we consistently observed this phenomenon in this type of experimental setup (confocal tracking of Alexa Fluor dyes). To directly show concentration-dependent multimerization of GAF-FL, and the monomeric state of GAF Δ POZ, we have added newly obtained mass photometry data (EDF 9d).

7. Line 316: when comparing diffusion coefficients of monomeric and multimeric GAFs, it would be useful to mention that a substantial difference in diffusion coefficients is expected purely based on the difference in size.

We thank the reviewer for this suggestion. We have modified the main text (lines 303-308) to mention that we expect at maximum a 4-fold difference in diffusion coefficients based on the difference in size between monomeric GAF Δ POZ (~80 kDa) and GAF-FL multimer (~800 kDa, based on mass spectrometry data added in Extended Data Fig. 9f). This cannot fully account for the observed 31-fold difference and strengthens our conclusion that multimerization is important for specific binding.

8. Line 298: given the limited resolution of tracking, it would be hard to distinguish short-range 1D diffusion from direct 3D binding. This point is mentioned in the discussion, but it would make sense to mention it here too and state the resolution of tracking.

We thank the reviewer for this suggestion and have modified the text accordingly (lines 294-296).

9. Some methods sections are missing intended references, having (Ref) marks instead, e.g., lines 891, 950, 952, and 962.

We thank the reviewer for noticing this and have inserted the missing references.

10. EDF 9: typos in panels b and e ('photobleaching').

We thank the reviewer for this suggestion and have corrected the typos.

11. EDF 4 and 5: it seems excessive to provide percentages up to the first decimal place, especially for cases with $N < 100$.

We thank the reviewer for this suggestion and have rounded the percentages to the nearest whole number for cases with $N < 100$.

12. Line 43: should probably read "...transcription factors such as LacI SEARCH efficiently for THEIR TARGETS..."

We thank the reviewer for this suggestion and have made the suggested changes.

Reviewer #2 (Remarks to the Author):

The manuscript by Feng et al. reports single-molecule fluorescence studies of the *Drosophila* transcription factor, GAGA-Associated Factor (GAF) interacting with DNA and nucleosomes containing GAF target sequences. Native full-length GAF (GAF-FL) functions as a multimer with an average of 6 monomers. While the authors include

studies of GAF-FL, they largely focus on a GAF mutant where much of the protein is deleted so it contains the minimal length for site-specific DNA binding. They refer to this protein, which functions as a monomer, as the GAF DNA binding domain (GAF-DBD). In addition, they study GAF with the POZ domain deleted, which abolishes multimerization. This results in GAF functioning as a monomer, which they refer to this construct as GAF-POZ.

Most of the measurements in the manuscript study fluorophore-labeled GAF-DBD with single molecule Total Internal Reflection Microscopy (smTIRF). They label GAF-DBD with Cy3 so that it undergoes efficient FRET with a fluorophore attached just outside the DNA target site. They studied both one site using Cy5, and two sites using Cy5 and Cy7. This elegantly allows them to observe both GAF-DBD binding to the DNA and when GAF-DBD is at one of the binding sites. They use different DNA constructs. One contains part of the native sequence from hsp70 promoter and another contains the Widom 601 high-affinity nucleosome positioning sequence. They carried out smTIRF measurements with a range of different constructs. Their main conclusion is that (i) GAF-DBD undergoes a combination of 1D and 3D diffusion to bind its target sequence and if there are two sites it slides back and forth along the DNA between the two sites, and (ii) GAF-DBD 1D diffusion is blocked by nucleosomes so that it relies on 3D diffusion to access sites within the nucleosome.

The authors also provide gel shift data to show that GAF-FL can bind to its target sequence within nucleosomes similar to free DNA. This supports the conclusion that GAF-FL functions as a pioneer factor. Finally, the authors provide C-Trap measurements of GAL-FL sliding along DNA tethers with multiple target sequences spaced semi-randomly with kb spacing. These results show GAF-FL can slide over kb distances and target multiple sites through 1D diffusion.

This is an extensive single-molecule study that address an important question, namely how do site-specific DNA binding transcription factors find their target sequences within chromatin? Their approach of using a combination of colocalization and FRET to determine both DNA binding and binding to a target sequence within the DNA molecule is elegant and insightful. These are the central findings of the work. The gel shift assay shows GAF-FL can target its site efficiently within nucleosomes and the C-Trap measurements on long DNA molecules provide additional interesting information. However, these last two measurements are not that well connected to the previous studies of GAL-DBD.

This is an interesting study with a significant amount of high-quality single-molecule (smTIRF) measurements of GAF. In addition, the study includes some gel shift and C-Trap measurements. But they are a bit disconnected from the smTRIF measurements, which causes the study to be a bit disjointed. However, overall, these studies provide important insight into how GAF targets sites within DNA and how nucleosomes influence monomeric GAF.

Below I provide feedback to help improve this study.

1. The authors report binding rates for the smTIRF and C-Trap measurements. However, they do not report (at least I could not find) the concentration of the GAF protein. The binding rates depend on the concentration, which is why one typically reports a binding rate constant in units of $\text{time}^{-1}\text{concentration}^{-1}$. The authors should include the GAF concentrations used in each experiment and include the binding rate constants in addition to the binding rates. Also, ideally, binding measurements are done at different concentrations to show the expected linear dependence of binding rate on concentration.

We thank the reviewer for the suggestions. Protein concentrations are stated in methods, but we have now added them to the main text and figures captions. We also measured binding rate as a function of GAF-DBD concentration and observed the expected linear dependency (Extended Data Fig. 2c).

2. Related to point 1, the authors do not provide ensemble measurements of the apparent K_D . This could be measured by electromobility gel shift assays (EMSAs) or by fluorescence. I assume they did these measurements to confirm the activity of their GAF samples. I strongly suggest they include EMSA measurements of GAF-DBD and GAF-POZ.

We thank the reviewer for this suggestion. We have now included EMSA measurements of GAF-DBD and GAF Δ POZ in Extended Data Fig. 2b and Extended Data Fig. 7e.

3. Related to point 1 and point 2, it would be good to compare the rate measurements to the apparent K_D of GAF-DBD. The authors just need to divide the binding rate of GAF-DBD by the concentration used to determine the binding rate constant. They can then divide the dissociation rate by the binding rate constant and compare this to the apparent K_D . These do not necessarily need to be the same, but they should be within the same order of magnitude.

We thank the reviewer for this suggestion and have performed the calculations. The association rate constant k_{on} of GAF-DBD, determined from a linear fit to the binding frequency as a function of protein concentration, is $0.27 \text{ s}^{-1} \text{ nM}^{-1}$. To calculate the K_D , we divide our measured binding frequency (3.1 s^{-1} , Fig. 1) by this number to get 11.5 nM . This agrees well with the 12 nM apparent K_d ($K_{1/2}$) determined from our GAF-DBD EMSA measurement (Extended Data Fig. 2b). We have included these calculations in the legend for Extended Data Fig. 2c.

4. The authors provide the cumulative distribution functions (CDF) for some of their dwell time measurements (Figures 4 and 5) but not for others. The Dwell times reported in Figures 1, 2, and 3 most likely come from CDFs. The authors should provide the CDFs for each effective dwell time that is determined from a distribution of dwell times with the fits used to determine the overall dwell time.

We thank the reviewer for this suggestion and have added 1-CDF plots with fits for dwell times reported in Fig 1-3 (for Fig. 1: Extended Data Fig. 2e; for Fig. 2-3: Supplementary Note)

5. There is a large number of different dwell times and rates reported. It is hard to keep track of all of them and to see how they compare. To help, the authors should provide a supplementary table with all the determined rates/dwell times with their uncertainty.

We thank the reviewer for this suggestion. A table with determined dwell times and rates with their uncertainty is now included in Supplemental Information.

6. It is interesting that the binding rate to DNA with the GAF-DBD target site (cognate DNA) is higher than to DNA without the target site (non-cognate DNA). This suggests that there are binding events to the non-cognate DNA that are too short to observe since it is unlikely that when GAF-DBD is not bound to DNA that it “knows” if the DNA has a binding site or not. The authors should discuss this more in the Results and/or Discussion sections.

We thank the reviewer for this suggestion and have discussed this technical limitation in more detail (lines 122-125).

7. It is challenging to keep track of the different sites and the distance between sites. It would be very helpful to provide names of the binding sites, to indicate them in the figure diagrams, and to provide the base pair spacing within the diagrams. This should be done by adding a diagram in Figure 1 and adding this information to Figures 2A, 3A, and 4A.

We thank the reviewer for this suggestion and have made the suggested changes to Figures 1b, 2a, 3a and 4a.

8. Similar to point 7, the authors should provide diagrams of the three protein constructs they use: GAF-DBD, GAF-FL, and GAF-POZ.

The diagrams of the three protein constructs are provided in Extended Data Fig. 2d.

9. For the 3-color FRET measurements in Figure 4, there could be two separate GAF-DBD binding to each site. The traces that show the Cy5 and Cy5 fluorescence increase simultaneously are very likely one GAF-DBD. However, there are likely traces with 2 GAF-DBD bound. The authors should be clearer about whether there are events with 2 GAF-DBD bound? If so, what is the fraction of these events? Alternatively, if not, then why are these not observed?

Traces with 2 GAF-DBD molecules bound are difficult to interpret because FRET with the acceptor fluorophores can come from either molecule. Thus we minimized such traces by using a low GAF-DBD concentration.

Given the low concentration we use, it is unlikely for two GAF-DBD molecules to arrive at the same time. Even if they arrived at the same time, their photobleaching times will be different, which would allow us to distinguish between single GAF-DBD binding. All binding events shown in Figure 4 are characteristic of single GAF-DBD binding. We have added this clarification to Methods: 3-color imaging of Cy3-GAF-DBD binding to Cy5 & Cy7-labeled DNA or nucleosome and Methods: 3-color data analysis.

10. It is interesting and perhaps surprising that the overall dwell time on the DNA fragment with 2 sites is similar to the dwell time on the 90 bp DNA with one binding site. Perhaps this is because the second site has a lower affinity sequence. The authors use a DNA construct later with 2 high-affinity sites. But they only study this DNA construct wrapped into nucleosomes. Did the authors happen to do DNA-only studies with the DNA construct with two high-affinity sites? It would be interesting to determine if the resident time is significantly increased.

We thank the reviewer for this comment. The DNA construct for the nucleosome study (in Fig. 4) is actually the same as the “flipped” DNA construct in Extended Data Fig. 4f. We have now stated this explicitly in Fig. 4 caption (lines 652-653). As stated in lines 157-159, the answer to the reviewer’s question is that we did not observe a significant increase in residence time when DNA contains two high-affinity sites (3.6 s) as opposed to only one site (3.1 s). One potential explanation is that, for GAF-DBD, the overall residence time on DNA is actually driven by dissociation from nonspecific sites into solution, which should be independent of the number of specific sites on the DNA. The number of high-affinity sites is more important for GAF-FL because of its multimerization capability. Indeed, we found that GAF-FL multimers stay at least two times longer on a DNA with clusters of high- and low- affinity sites than on DNA with only isolated sites (Fig. 5j).

11. The authors report where the binding site within the nucleosome is located relative to the superhelical location (SHL). Is this based on the center of the binding site? It would be helpful if the authors also reported how far the binding site extends into the nucleosomes.

Yes, the superhelical locations are based on the center of the binding sites. For example, the SHL7 site replaces base pairs 2-7 on the 147 bp 601 sequence (67-72 bp from the dyad); the SHL5 site replaces base pairs 22-27, etc. We included details on the DNA sequence in Supplemental Information.

12. For the measurements with nucleosomes, how do the authors know that their molecules contain a full histone octamer after surface tethering? Is it possible that some of the molecules have lost one or both histone heterodimers? The authors should discuss this possibility.

If our nucleosomes lost one or two heterodimers after surface tethering, then the binding site at SHL6.5 should be accessible to 1D diffusion. Since we observed a

near-complete inhibition of 1D diffusion at SHL6.5, the evidence indicates that our nucleosomes remain intact after surface tethering.

13. The authors use abbreviations that are not clearly defined in the manuscript (at least I could not find definitions). This includes CDF, which is a cumulative distribution function. Also, the authors refer to A-A, B-B, and C-C events, but these events do not appear to be defined. Please define all abbreviations in the manuscript.

We thank the reviewer for this suggestion and have added the definitions accordingly.

14. The authors provide SDS PAGE gels of the different GAF protein samples. However, there is no characterization of the reconstituted nucleosomes. Typically, this is done with Electromobility Shift Assays (EMSAs). The authors should provide EMSAs of the different nucleosome constructs used in the different single-molecule assays.

We thank the reviewer for this suggestion and have added nucleosome EMSAs to Extended Data Fig. 5a.

15. The methods references have issues that need to be addressed. In some places, the text has (Ref). I assume this is a placeholder and the authors forgot to insert the correct reference. In other places, the authors list the last name of the first author and the year. Each reference should have a full reference that is listed at the end of the methods.

We thank the reviewer for this reminder and have inserted the missing references.

16. More details in the Methods section need to be provided. They provide excellent detail on the preparations of GAF-DBD and GAF-DBD. However, details on the histone octamer and nucleosome preparations are mostly lacking. The authors should not completely rely on referencing previous papers. Please provide in the Methods section more details on how the histone octamer was refolding and purified, and how nucleosomes were reconstituted and purified.

We thank the reviewer for this suggestion and have inserted detailed methods for the purification of histones, histone octamer, and nucleosomes.

Reviewer #3 (Remarks to the Author):

Feng et al. present a single molecule study of GAGA-associated factor (GAF) binding and diffusion on DNA and nucleosomal templates. Both the DNA binding domain and the full-length GAF can engage in 1D diffusion on naked DNA, which facilitates target search. Nucleosome blocks sliding beyond the entry / exit sites, and solvent-exposed motifs on the nucleosomes are mostly accessed through direct binding.

While the findings may not be groundbreaking, this paper offers a valuable contribution to the field. GAF plays a crucial role in gene regulation and development, particularly as a pioneer factor, making its interaction with nucleosomes highly relevant from a

physiological perspective. The use of both the 601 nucleosome positioning sequence and endogenous GAF binding sequence demonstrates a rigorous approach. The diffusion data obtained through 3-color FRET and C-trap methods are particularly intriguing.

There are several points that require further clarification:

1. Is the data in Figure 1 measured at one concentration of GAF-DBD? If yes, how is this concentration determined? Does the single molecule data roughly agree with bulk measurement of K_D ?

We have now included binding rates at more concentrations in Extended Data Fig. 2c as well as bulk EMSA data in Extended Data Fig. 2b. K_D determined from single-molecule data (11.5 nM) agrees well with the bulk K_D (12 nM). We have discussed this in detail in the legend for Extended Data Fig. 2c.

2. Figure 1g & h. The error bars for the "technical replicates" appear large. How were these calculated? The number of traces and binding events should be reported for accuracy.

The binding frequency error is the standard error of the mean (S.E.M.) for binding frequencies from multiple technical replicates. Each replicate contains more than 100 single-molecule traces.

The reported dwell time error is the S.E.M. of τ 's obtained by fitting a single-component exponential decay function to 1-CDFs. Each 1-CDF is constructed from more than 100 dwell times from one replicate.

We have added the details above to Methods: 2-color smFRET data analysis. We have also included 1-CDF plots with their fits in Extended Data Fig. 3e.

3. The average dwell time of GAF in Figure 1 seems significantly longer than that on Sites 1 and 2 in Figure 2, despite Site 2 having the same motif. What explains this difference? If it's due to entrapment at Site 1, what is the average dwell time on Site 2 after mutating Site 1 (see Extended Data Figure 4c-e)?

The average dwell time on DNA (from binding to DNA to dissociation into solution) is similar between the Fig. 1 construct (3.1 s, cognate DNA construct) and the Fig. 2 construct (3.34 s). The motif dwell time (excluding the time protein spends 1D sliding or dwelling on non-specific DNA) in Fig. 1 does appear longer than Site 2 in Figure 2, which we assume is what the reviewer is referring to. We do not interpret this as entrapment at Site 1 or other context-dependent protein behavior, mainly because the Fig. 1 single-site construct is not the best construct for resolving the motif dwell time. If GAF-DBD slides off the cognate site and quickly returns within exposure time, we would continue to observe a Cy5 FRET signal, whereas with the Fig. 2 Cy5&Cy7 construct it would likely slide to the other site and undergo FRET with the other acceptor fluorophore, allowing us to resolve motif dissociation. In addition, the camera exposure time used for two-color imaging (100 ms) was longer than that used for three-color imaging (35 ms), and that the non-motif part of the DNA was longer for Cy5&Cy7 constructs (83 bp in Fig. 1 vs 180 bp after mutating Site 1 in Extended Data Figure

4c-e). All these factors might have contributed to missing sliding events in the two-color traces.

4. Figure 3 only presents the binding site oriented towards the histone. It would be beneficial to show some traces of SHL2.5, 4.5, and 6.5, ideally also the corresponding rastergrams. The extensive scanning at SHL4.5 (significantly more than SHL6.5 according to Extended Data Figure 5) is surprising and warrants an explanation.

Trace examples have been added for 601-SHL2.5, 4.5 and 6.5 nucleosomes in Extended Data Fig. 5d.

“Scanning events” are binding events that show dynamic Cy3-Cy7 FRET within a binding event as opposed to “nucl. binding events” where Cy3-Cy7 FRET spans the entire binding duration. A scanning event might come from (1) free DNA contamination in the sample (as seen in SHL5 and SHL3 data), (2) diffusion from the linker site to the internal site (Figure 3h “Linker & Nucl.” events), or (3) diffusion from a nonspecific site to the internal cognate site. Native PAGE characterization of the SHL4.5 nucleosome suggested there was a small amount (~10%) of contaminating free DNA (Extended Data Fig. 5a), but the level was similar to SHL2.5 and SHL6.5 nucleosomes. Thus free DNA contamination cannot explain the increased scanning at SHL4.5. We think most scanning events on 601-SHL4.5 nucleosomes were likely type (2) or (3) because we observed binding events where the Cy3-Cy7 FRET duration was longer than Cy3-Cy5 FRET (Extended Data Fig. 5d, right) and Cy3-only. This suggests a possibility for GAF-DBD to hop from either a cognate site on the linker or a nonspecific site to an internal cognate site, although the nature of the preference for SHL4.5 remains unclear especially since it is spatially farthest from the nucleosome edge. We have discussed the above in the figure legend for Extended Data Fig. 5.

5. I am not fully convinced of the data interpretation in Figure 4C. The diagram in the middle (dyad site distal to Cy7) should lead to decreased FRET efficiency of Cy5 and Cy7, but the traces above do not show such trend. Some explanation should be provided. Maybe the Cy7 motif can be shortened to eliminate the possibility of multi-mode binding.

We thank the reviewer for this comment. We agree that the data in Fig. 4e is not fully explained by the nucleosome model below because one would expect a higher Cy7 FRET signal than the Fig. 4d scenario if GAF-DBD binds closer to the Cy7 fluorophore. We have adjusted our interpretation of these events based on the following observation: binding events showing Cy3-Cy7 FRET only contribute less than 13% of binding events yet once it occurs, the next binding event is likely to also show Cy3-Cy7 FRET only (Extended Data Fig. 6a bottom trace, Extended Data Fig. 6b, right). This indicates that the hsp70 DNA may adopt a certain phase around the nucleosome to favor a binding site that only results in ~0.24 FRET efficiency between Cy3 and Cy7. Conversion from FRET efficiency to distance suggests that Cy3-GAF-DBD is 4-5 nm from Cy7 and at least 9 nm from Cy5. The distance from Cy7 can be explained by GAF-DBD binding at the distal end of the Cy7-Site 1 (15 bp on the nucleosome ~ 4-5 nm from the Cy7

fluorophore) but the longer distance from Cy5 can only be explained by a different DNA wrapping from the dominant phase perhaps such that Cy5 is outside or at the breathable edge of the nucleosome. This is supported by the a newly added Cy5-Cy7 FRET histogram for these nucleosomes showing a dominant Cy5-Cy7 FRET peak near zero as well as a significant spread over non-zero FRET efficiencies up to 0.4 (Extended Data Fig. 6c). In summary, we are still confident in assigning binding events showing Cy3-Cy7 FRET only to the internal site, but the nucleosome phasing is potentially distinct from the Cy5 & Cy7 dual FRET category. We have modified lines 237-241, Fig. 4 and Extended Data Fig. 6 to reflect this interpretation.

6. Related to question 3 above, the dwell time on the same motif but in different templates seem to vary significantly. I thus wonder if the comparison in Figure 4g is fair. How about the dwell time of GAF on 601 nucleosome at SHL2.5, 4.5, and 6.5? Do they also show longer dwell time?

We think it is reasonable to compare the dwell times as in Fig. 4g because the same DNA template was used for the free DNA and the nucleosome samples. On the second question, the answer is yes, average dwell times of GAF on 601 nucleosome at SHL2.5, 4.5, and 6.5 are 0.82 s, 0.90 s, and 0.87 s, respectively, which is consistent with the dwell time being longer than on naked DNA (0.11 s when a motif of a similar length, Cy5-Site 2, is on naked DNA). We have added this result to lines 259-261.

7. Current Figure 4 is somewhat confusing because the data in panels a-g are based on GAF DBD, whereas panel h uses the full-length GAF. Including an EMSA comparison between nucleosome and naked DNA using GAF DBD would enhance clarity.

We thank the reviewer for this suggestion. EMSA for GAF-DBD binding to hsp70 nucleosome and DNA have been added to Fig. 4h and Extended Data Fig. 2b.

Minor:

In Figure 2c, the diagram appears to indicate "50" rather than "55".

It would be more informative to show 1-CDF using semi-log plot.

Figure 4h is not cited correctly in the manuscript; it is referred to as Extended Data Figure 6 instead.

We thank the reviewer for these suggestions. We understand that 1-CDFs are sometimes presented in the semi-log format to better visualize the difference in data especially in the lower x-axis. In our case, we already see clear differences in the unconverted format, so reformatting to semi-log plots is unnecessary and will not affect our conclusions. For this reason we have kept the 1-CDF plots the same. This is also to stay consistent with our newly added 1-CDF plots for smFRET dwell times (Extended Data Fig. 3e and Supplementary Note). The other two minor suggestions have been resolved.

Reviewer #4 (Remarks to the Author):

This manuscript presents significant scientific findings regarding how eukaryotic sequence-specific transcription factors (TFs) search for gene targets on chromatin, influenced by nucleosomes. The research employs state-of-the-art approaches, making the discoveries relevant for publication in NSMB. I generally support publication but have suggestions for improving the manuscript's quality and clarity.

1. Clarification of 1D vs. 3D Search: The physiological significance of 1D or 3D search modes remains unclear. The authors need to explain why this behavior is important. It would be helpful to focus on the observations in the model shown in Extended Data Figure 10. Speculation should be minimized. The authors should emphasize the data and clarify the functional differences between 1D and 3D searching mechanisms.

We thank the reviewer for this suggestion. We have now elaborated on the physiological significance of 1D and 3D search modes as well as emphasized how our data furthered our understanding of TF target search in the discussion (lines 318-337, 397-403).

2. Multimeric Status of the TF: The multimeric status of the transcription factor requires more clarity. Although the authors suggest an average hexameric state, the data in Extended Data Figure 9d do not convincingly support six photobleaching steps. Moreover, the intensity differences between Extended Data Figure 9b and 9d are quite large although the intensity unit is arbitrary. I recommend comparing the intensities of a single spot with single photobleaching step intensities for a more quantitative analysis. If there are no structural studies supporting the hexameric state, the possibility that the multimeric status is a condensate should be considered, as many TFs display condensation properties. A more quantitative description is needed to clarify the function of this multimeric state.

We thank the reviewer for this suggestion. We agree with the reviewer that quantifying the exact number of photobleaching steps from C-Trap traces (Extended Data Fig. 9b and 9d) is challenging given the signal-to-noise ratio and variable single-fluorophore intensities from experiment to experiment, especially given that GAF-FL and GAF- Δ POZ were labeled with different fluorophores. Therefore, the intensity number between experiments is not directly comparable and explains the large intensity difference between extended data figures 9b and 9d. The tracked 488 traces also include a higher amount of background signal. We have now quantified the multimerization state of the same sample used in the optical tweezers experiments using mass photometry. The results show GAF multimerization is concentration dependent and largely hexameric to octameric in the same concentration used in the C-Trap imaging conditions. Notably at 50nM Δ POZ no higher molecular weight products were observed. We have added the mass photometry data to Extended Data Fig. 9f. The new mass photometry data directly show that GAF-FL forms a range of multimers via its POZ domain. We think this experiment addresses the reviewer's concern and provides novel insight into GAF's multimerization property. We have discussed this in lines 299-308.

3. 1D Diffusion Mechanism: It would be helpful if the authors structurally explain how the GAF DBD or its multimeric state can perform 1D diffusion. Does GAF DBD possess a DNA anchoring site that enables it to slide along DNA? Alternatively, could there be a pore created by the assembly of GAF protomers?

GAF-DBD consists of a single zinc finger domain (ZnF) flanked by three basic regions (BRs), two on the N terminal side and one on the C terminal side. A previous structural study showed that the ZnF and the N-terminal BRs make base-specific contacts with the major and minor grooves as well as the sugar phosphate backbone (PMID 9033593). The backbone contacts could allow nonspecific scanning of the DNA. We have added this to the discussion (lines 322-326).

4. Hopping vs 1D diffusion in FRET Data: The authors should clarify whether the FRET results exclude the possibility of hopping events, which are commonly observed during DNA-protein interactions. If hopping is not occurring, the authors should rule out this possibility in the manuscript. Additionally, for 1D diffusion, a gradual change in FRET values would be expected. Please provide this data.

Proteins undergo 1D diffusion on DNA via either hopping (with transient and short-range dissociation from DNA followed by re-binding to a nearby site) or rotation-coupled sliding along the DNA double helix. (PMID 18544605; DOI 10.1088/1751-8113/42/43/434013) Our data cannot directly distinguish between the two mechanisms and the 1D movements we observed can be a mix of the two behaviors as long as the hopping occurs close enough to the slide surface (within the TIRF evanescent field depth ~ 100 nm) and to the substrate molecule (within diffraction limit ~ 200 nm). One way to experimentally distinguish between hopping and helically coupled sliding is by changing the ionic strength of the buffer: if hopping is involved, 1D diffusion should be slower at a lower ionic strength due to tighter association with the DNA, whereas helically coupled sliding should be unaffected. (PMID 18544605, 17996701) We tested different ionic strengths of the buffer by changing either the NaCl or MgCl₂ concentration and found that the nonspecific state (exhibiting Cy3-only fluorescence) becomes significantly longer at lower ionic strengths (Extended Data Fig. 3). This suggests that the 1D diffusion of GAF-DBD involves some hopping along the DNA (lines 112-114).

The reviewer mentioned that a gradual change in FRET efficiency would be expected in case of (helically-coupled) 1D diffusion. While this is a reasonable guess given that a gradual departure from one cognate site and a gradual approach to the other site should give rise to a monotonic shift in FRET between the fluorophores, we actually do not expect to detect this shift under our time resolution (~ 50 ms) based on the measured diffusion coefficient of monomeric GAF (Fig. 5k, Δ POZ, $0.22 \mu\text{m}^2/\text{s}$). If we were to detect such a gradual shift that spans three time steps, for example, a calculation based on the equation $MSD = 2Dt$ suggests that the diffusion coefficient should be within the order of magnitude of $0.001 \mu\text{m}^2/\text{s}$. This is two orders of magnitude smaller than the expected diffusion coefficient of GAF-DBD based on the measurement for GAF Δ POZ even after accounting for their size difference (the molecular weight of GAF Δ POZ is 3.5 times that of GAF-DBD which suggests the diffusion coefficient of

GAF-DBD is at most 2 times that of GAFΔPOZ based on calculations similar to Methods: Estimation of theoretical diffusion coefficients). This means that we expect transitions between the two cognate motifs to occur much faster than if they were to appear as a gradual change in FRET and instead appear as direct transitions between Cy5 and Cy7 signals. Therefore we do not expect to detect a gradual change in FRET efficiency for GAF-DBD 1D diffusion.

5. Cy5 and Cy7 Data Interpretation: In the Cy5 and Cy7 data, where the motif is positioned at the nucleosome dyad and edge, there are instances where only one signal lights up and cases where both signals light up simultaneously. This raises the question of whether there are three distinct binding modes or if the wide variation in Cy5 to Cy7 signal ratios suggests a more dynamic binding mode. The FRET values should be provided to clarify whether the GAF-DBD bound state is a single state or multiple intermediates.

We thank the reviewer for raising this question. While our hsp70 nucleosome adopts one dominant position as shown by native PAGE and Cy5-Cy7 FRET analysis (FRET histogram now provided in Extended Data Fig. 6d), we think this nucleosome can actually adopt at least three distinct phases (most likely less than 5-bp shift between each phase) that can neither be separated by PAGE gel—which is less sensitive to movements smaller than 10 bp (PMID 15262970, 30979890)—nor by Cy5-Cy7 FRET—since both fluorophores are on DNA, a shift in nucleosome position does not necessarily change their relative distance to change their FRET efficiency. Our observation that GAF-DBD “prefers” to re-bind the same position (Extended Data Fig. 6b,c) supports this idea and suggests that each phasing may present a specific GAGAG site for optimal binding. FRET values for Cy3/Cy5 and Cy3/Cy7 pairs are not shown because we found it more informative to focus on categorizing the binding events based on the presence or absence of their FRET signal (Fig. 4f) instead of FRET efficiency values. This is because to further assign binding positions beyond edge/dyad binding based on the FRET values, we need to link these FRET values to nucleosome positions at a single-nucleosome level with base-pair accuracy which is beyond the scope of this paper. We have accordingly modified Fig. 4, the figure legend of Extended Data Fig. 6 and the text (lines 237-241) to clarify this.

The FRET values are provided here for the reviewer’s reference:

Extended Data Fig. 6a	Cy3-Cy5*	Cy3-Cy7*	Cy5-Cy7#
Top trace	0.45 ± 0.08	0.07 ± 0.1	0.01 ± 0.03
Middle trace	0.72 ± 0.14	0.51 ± 0.17	0.22 ± 0.02
Bottom trace	0.03 ± 0.08	0.24 ± 0.08	0.27 ± 0.09

** averaged from FRET values during binding*

averaged from direct Cy5 excitation at the beginning of imaging

6. Transition Between Cy5 and Cy7 Signals: The manuscript shows a Cy3-signal recovery during transitions between Cy5 and Cy7 signals, which suggests sliding between two binding sites on DNA. However, there are cases where Cy5 and Cy7 transitions occur directly, which may not indicate sliding. Do the authors have statistical data on the ratio of direct transitions versus Cy3-mediated transitions? If there are missed Cy3 detections during Cy5 to Cy7 transitions, it would be helpful to address this limitation.

We thank the reviewer for this suggestion. As we have mentioned in response to Question #4, we do believe that most “direct” transitions from Cy5 to Cy7 signal are fast 1D diffusion events with an undetectable Cy3-only signal in between. We have addressed this limitation as well as reported the ratio of between “direct” transitions and Cy3-mediated transitions on lines 153-156.

7. Conversion of Length to Base Pairs: The conversion between length and base pairs may vary depending on the force applied in the optical tweezer experiments. Was this calculation based on an assumed value of 0.34 nm per base pair? Additionally, the authors should clarify if the binding between DNA and protein changes depending on the tension applied to the DNA.

The optical tweezer measurements were obtained under minimal tension of 5pN. Assumed base pair value of 0.34nm was used to estimate distance to base pair conversion. Stretching and tether orientation can influence the precise accuracy of this estimate. However, empirically we know this to be in the range of distances observed for a known fragment of double stranded 48.5 kb lambda DNA under 5pN (PMID 35876491). It is possible that tension applied to the DNA can influence GAF protein binding however these differences were not characterized in this study.

8. Kymographs in Figure 5c,d: The kymographs in Figure 5c,d show changes in protein intensities. The authors need to clarify whether these changes are due to photobleaching or the dissociation of multimeric states.

The gradual decline of fluorescence of all the fluorophores is most consistent with previous characterizations of photobleaching by confocal scanning on optical tweezers (PMID 37828742). However, we cannot rule out changes in the multimerization state especially when sliding. It seems unlikely that there is significant dissociation of multimeric states given the absence of a monomer peak in the mass photometry data at the concentration range that was used in the imaging experiments (Extended Data Fig. 9f).

9. Cy3 Donor-only Fluorescence: In line 112 on page 3, the authors should clarify that the transient Cy3 donor-only fluorescence flanked by FRET events is not a result of photo-blinking.

We thank the reviewer for this suggestion. The photo-induced blinking of Cy5 fluorophore is negligible under our imaging conditions. If one directly excites Cy5, we would observe a continuous Cy5 emission signal (see example in Poyton & Feng et al Supplementary Fig. 1 PMID 35263135). Our ability to modulate the FRET dynamics by changing salt concentration also suggests that the dynamics is not due to photo-induced blinking which should be more dependent on laser intensities and imaging buffer than on salt concentration. We have clarified this in lines 713-714.

10. Binding Time and 1D Sliding: In the early section, the manuscript mentions that GAF-DBD remains bound to the motif about 2% longer than the theoretical binding time calculated based on 3D diffusion. If the theoretical 3D binding probability is 66%, does this mean that 60 bp of the 90 bp DNA used in the experiment consists of the GAGAG motif, or is this a thermodynamic probability based on binding energy in 3D? Clarification is needed.

We assume the reviewer was referring to line 100 where we discussed the theoretical probability for a 3D collision between GAF-DBD and DNA to occur at the cognate binding site. The probability is calculated based on the number of available cognate and noncognate binding sites. For example, given a 10-bp DNA CTGAGAGAGT—which contains 2 possible GAGAG sites and 4 possible non-cognate binding sites—the probability of a successful collision is $2/(2+4) = 33.3\%$. We have clarified this in lines 101-102.

11. Rebinding Tendency: In Extended Data Figure 6, it is noted that dissociation and re-binding of the same type (A-A, B-B, C-C) occur more than 80% of the time. Could this observation be influenced by nucleosome assembly differences? The authors should clarify whether similar trends are observed with longer interval times before re-binding.

We think the nucleosome-to-nucleosome heterogeneity in binding positions is likely due to our use of the non-601 native DNA sequences which may result in subtle differences in nucleosome positioning and differentially presented cognate sites. Only one band was observed on the native PAGE gel for the reconstituted nucleosome (Extended Data Fig. 5a, hsp70 NCP), suggesting that nucleosomes are properly assembled.

As shown in Extended Data Fig. 6a, the same position was repeatedly visited throughout a 90-second time window, with up to 25-second intervals between binding. We have not tested longer imaging times (to achieve longer interval times) as our Cy5 and Cy7 fluorophores would likely photobleach in the middle of imaging and make it difficult to interpret later binding events.

12. 1-CDF Data Comparison: The graph calculated with 1-CDF shows that motif-specific dwell time differs tenfold between DNA and nucleosomes. Is this comparison based on maximum values, or does it refer to a comparison of average data? Clarification would improve interpretation.

It is based on the τ in a single-exponential fit to 1-CDFs, which is close to the mean. We have included the fitted values in Supplemental Information Table 5 and clarified this in the text (lines 255-256) and Fig. 4e legend.

13. GAF-FL Binding Comparison: In line 276, the manuscript compares GAF-FL binding to NCP versus bare DNA via titration, noting that NCP binds about half as much as bare DNA. Could the calling in Extended Data Figure 6 be double-checked to ensure accuracy?

We thank the reviewer for pointing this out. We have corrected the figure citation to Figure 4h.

14. Typo: In line 307, please delete “, d”.

We have corrected this typo.

I believe that addressing these points will significantly strengthen the manuscript's quality and scientific rigor.